# Reinforcement Learning with Pairwise Preferences in Long-Term Decision Problems

**Jonathan Colaço Carr**[* 1 2]  **Prakash Panangaden**[1 2]  **Doina Precup**[1 2]  **Benjamin Van Roy**[3]

## Abstract

Reinforcement learning problems typically define the goal as maximizing the expected value of a scalar reward function. But, pairwise preferences are often easier to specify than scalar rewards, and they express certain goals that scalar rewards cannot. Methods for reinforcement learning with pairwise preferences have thus received growing interest. Unfortunately, these methods are inefficient in problems with long time horizons, and they lack guarantees on the performance of Markov policies relative to history-dependent policies, which bridge the theory and practice of reinforcement learning. We therefore propose the *Markov decision contest* as a new problem model for reinforcement learning with pairwise preferences. We prove that stationary Markov policies are optimal among all history-dependent policies, that solving a Markov decision contest exactly is in P, and that a simple iterative algorithm converges to an optimal policy at a sublinear rate. Lastly, in a set of high-dimensional decision problems with long time horizons, we show that our approximate algorithm is significantly more learning-efficient than prior work.

## 1. Introduction

Traditional reinforcement learning problems are formulated as Markov decision processes (Puterman, 2014; Sutton & Barto, 2018), in which the goal is to maximize the expected value of a scalar reward function. But, pairwise preferences are often easier to specify than scalar rewards. So, for many applications, such as fine-tuning large language models (Stiennon et al., 2020) the problem begins with pairwise preferences, instead of a reward function.

Reinforcement learning from human feedback (Christiano et al., 2017; Casper et al., 2023) has tried to resolve this by inferring a reward function from pairwise preferences. It then trains a reinforcement learning algorithm to maximize the expected value of the inferred reward function. But, this approach has fundamental limitations: not every pairwise preference function can be represented as a reward function (Fishburn, 1982). So, for some preferences, no reward function can correctly capture the goal—and maximizing an inferred reward function may lead to the wrong behaviour.

One alternative is to design new reinforcement learning algorithms that learn directly from pairwise preferences, without needing a reward function. The objective for these algorithms is to find a policy at the Nash equilibrium of a pairwise-preference function, which can be interpreted as a policy that no other policy is consistently preferred to.

Current methods of reinforcement learning with pairwise preferences, though, have two key limitations that Markov decision processes do not. First, these methods rely heavily on the fact that the problem ends at a known, fixed timestep, which does not seem practical for applications like large language models (LLMs), where people don't end at a fixed time. These methods also lack guarantees on the performance of Markov policies relative to history-dependent ones. These are important in the theory of Markov decision processes, as they justify solution methods that restrict their focus to Markov policies.

We introduce the *Markov decision contest* to study reinforcement learning with pairwise preferences in long-term decision problems. We show three things about this model:

1. Stationary Markov policies are optimal within the set of all history-dependent policies (Section 5.2).

2. The problem of solving a Markov decision contest exactly is in P. That is, it can be done in time polynomial in the number of states and actions (Section 6).

3. An approximate solution method, called *Hedged Pol-*

---

[*]Work done while at Stanford University. [1]School of Computer Science, McGill University, Montreal, Quebec, Canada [2]Mila - Quebec AI Institute, Montreal, Quebec, Canada [3]Department of Electrical Engineering, Stanford University, Stanford, California, USA. Correspondence to: Jonathan Colaço Carr <jonathan.colacocarr@mail.mcgill.ca>.

*Proceedings of the 43$^{rd}$ International Conference on Machine Learning*, Seoul, South Korea. PMLR 306, 2026. Copyright 2026 by the author(s).

*icy Iteration (HPI)*, converges to an optimal policy at a sublinear rate and scales well to function approximation (Sections 7 and 8). Specifically, HPI converges at a rate of $1/\sqrt{K}$, where $K$ is the number of iterations. We validate its performance in thirteen Markov decision contests using techniques from deep reinforcement learning.

These results suggest that reinforcement learning with pairwise preferences in long-term decision problems is more tractable than previously thought. This is especially relevant for large language models, as they continue to be used in decision problems with longer time horizons.

## 2. Preliminaries

All quantities will be defined with respect to a decision-making *environment* $(\mathcal{S}, \mathcal{A}, P, \mu)$ that consists of: a finite set of states $\mathcal{S}$; a finite set of actions $\mathcal{A}$; a transition probability function $P : \mathcal{S} \times \mathcal{A} \to \mathrm{Dist}(\mathcal{S})$; and an initial state distribution $\mu \in \mathrm{Dist}(\mathcal{S})$.

After $t$ steps of interaction, the decision-maker, or *agent*, will have generated a *history* $h_t = (s_0, a_0, \ldots, s_{t-1}, a_{t-1}, s_t)$, which is a finite sequence of alternating states and actions that begins and ends with states. The set of all histories of length $t$ is denoted by $\mathcal{H}_t$. The probability that the agent transitions to state $s' \in \mathcal{S}$ given state $s \in \mathcal{S}$ and action $a \in \mathcal{A}$ is written as $P(s'|s, a)$.

The *horizon* of the decision problem is the number of interaction steps. The horizon can be either finite or infinite.

### 2.1. Decision Rules and Policies

We differentiate between history-dependent and Markov policies following Puterman (2014, Chapter 2). This notation differs from Sutton & Barto's, which may be more familiar to some readers.

A *decision rule* specifies which action to take at a given timestep $t$. The most general rule is a *randomized history-dependent decision rule* $d_t^{HR} : \mathcal{H}_t \to \mathrm{Dist}(\mathcal{A})$, which assigns action-selection probabilities that depend on all previous states and actions. A *randomized Markov decision rule* $d^{MR} : \mathcal{S} \to \mathrm{Dist}(\mathcal{A})$ selects action probabilities that depend only on the state at the current timestep. A *deterministic Markov decision rule* $d^{MD} : \mathcal{S} \to \mathrm{Dist}(\mathcal{A})$ is a randomized Markov decision rule that only assigns degenerate probability distributions.

**Definition 2.1.** A **policy** $\pi = (d_t)_{t=0}^{\infty}$ is an infinite sequence of decision rules. It is said to be:

1. **history-dependent and randomized (HR)** if, for all $t$, $d_t$ is a history-dependent decision rule.

2. **Markov and randomized (MR)** if, for all $t$, $d_t$ is a randomized Markov decision rule.

3. **stationary and randomized (SR)** if there is a randomized Markov decision rule $d^{MR} : \mathcal{S} \to \mathrm{Dist}(\mathcal{A})$ such that, for all $t$, $d_t = d^{MR}$.

4. **stationary and deterministic (SD)** if there is a deterministic Markov decision rule $d^{MD} : \mathcal{S} \to \mathrm{Dist}(\mathcal{A})$ such that, for all $t$, $d_t = d^{MD}$.

For a Markov decision rule $d^{MR}$, the policy **given by** $d^{MR}$ is the stationary randomized policy $(d^{MR})_{t=0}^{\infty}$. In this case, we say that $d^{\pi}$ **determines** $\pi$.

The sets of all HR, MR, SR, and SD policies are denoted by $\Pi^{HR}, \Pi^{MR}, \Pi^{SR}$, and $\Pi^{SD}$, respectively. Because every randomized Markov decision rule can be identified as a randomized history-dependent decision rule, these policy sets are related as follows:

$$\Pi^{HR} \supset \Pi^{MR} \supset \Pi^{SR} \supset \Pi^{SD}.$$

### 2.2. Reinforcement Learning with Pairwise Preferences

**Definition 2.2.** Given a finite set of outcomes $\mathcal{Z}$,

- A **pairwise-preference function** is a function $\mathcal{P} : \mathcal{Z} \times \mathcal{Z} \to [0, 1]$ that satisfies, for all outcomes $z, z' \in \mathcal{Z}$,

$$\mathcal{P}(z, z') = 1 - \mathcal{P}(z', z).$$

- A **preference margin** $M : \mathcal{Z} \times \mathcal{Z} \to \mathbb{R}$ is a function that satisfies, for all outcomes $z, z' \in \mathcal{Z}$,

$$M(z, z') = -M(z', z).$$

When $\mathcal{Z} = \mathcal{S} \times \mathcal{A}$, we will call $M$ a **Markov preference margin**.

There is a one-to-one relationship between pairwise-preference functions and preference margins. For every pairwise-preference function $\mathcal{P}$, the function $M_{\mathcal{P}}$ given by

$$M_{\mathcal{P}}(z, z') = \mathcal{P}(z, z') - 1/2,$$

is a preference margin. We use the term "preference margin" because pairwise preference functions are used in other fields of computer science (e.g Shah & Wainwright (2018); Vitelli et al. (2018)).

The number $M(z, z')$ represents the amount of utility that would be traded to observe $z$ rather than $z'$. Thus, $M(z, z') > 0$ if $z$ is preferred to $z'$, and $M(z, z') = 0$ if there is no preference (or, indifference). In Appendix D, we discuss how preference margins relate to the Bradley-Terry model of stochastic choice.

*Table 1.* Markov decision contests (MDCs) are a new problem model for reinforcement learning with pairwise preferences. They model decision problems with both finite and infinite horizons, and they have guarantees on the performance of stationary Markov policies relative to history-dependent policies. Surprisingly, MDCs are in the same complexity class as Markov decision processes.

| | Problem-Model Desiderata | | | Exact Methods | Approximate Methods | |
|---|---|---|---|---|---|---|
| | Pairwise | Horizon | $\pi^{SR}$ vs. $\pi^{HR}$ | Complexity | Convergence | FA |
| Gilbert et al. (2015; 2016) | ✓ | finite | ✗ | ? | ✓ | ✗ |
| Chen et al. (2022) | ✓ | finite | ✗ | ? | ✓ | ✗ |
| Wang et al. (2023) | ✓ | finite | ✗ | ? | ✓ | ✗ |
| Swamy et al. (2024) | ✓ | finite | ✗ | ? | ✓ | ∼ |
| Shani et al. (2024) | ✓ | finite | ✗ | ? | ✓ | ✓ |
| Wu et al. (2026) | ✓ | finite | ✗ | ? | ✓ | ✓ |
| Markov Decision Contests | ✓ | finite or $\infty$ | ✓ | poly$(|\mathcal{S}||\mathcal{A}|)$ | ✓ | ✓ |
| Markov Decision Processes | ✗ | finite or $\infty$ | ✓ | poly$(|\mathcal{S}||\mathcal{A}|)$ | ✓ | ✓ |

Problem models for reinforcement learning with pairwise preferences consist of a decision-making environment $(\mathcal{S}, \mathcal{A}, P, \mu)$ and a preference margin $M : \mathcal{Z} \times \mathcal{Z} \to \mathbb{R}$. Chen et al. (2022); Wang et al. (2023), and Swamy et al. (2024) consider the case where $\mathcal{Z} = \mathcal{H}_N$, where $N \in \mathbb{N}$ is a fixed, finite time horizon. The objective they consider is

$$\max_{\pi \in \Pi^{HR}} \min_{\pi' \in \Pi^{HR}} E_{\pi, \pi'}[M(H_t, H_t')].$$

Gilbert et al. (2015; 2016), and Shani et al. (2024) consider the case where $\mathcal{Z} = \mathcal{S}$, and where the objective depends only on preferences between states that occur at the final timestep $N$.

## 3. Related Work

Table 1 summarizes our contributions relative to prior work. We elaborate on related work in Appendix B. The first three columns are for desiderata that existing problem models lack, while the last three concern solution methods. The column properties are:

- *Pairwise:* whether the objective of the problem model is described by a pairwise preference function.

- *Horizon:* the horizon of the decision problem.

- $\pi^{MR}$ v.s. $\pi^{HR}$: whether the problem model has guarantees on the performance of Markov policies relative to history-dependent policies.

- *Complexity:* Computational complexity of solving the problem exactly.

- *Convergence:* Whether the problem admits approximate solution methods with convergence guarantees.

- *FA:* Whether the proposed approximate solution methods were validated in experiments with function approximation (FA).

## 4. Infinite-Horizon Decision Problems

The theory of infinite-horizon decision problems differs depending on whether the performance criterion is discounted or averaged. In Colaço Carr (2026, Chapter 4), we study a discounted version of the Markov decision contest. Here, we will study the average case. This will require the following material, which is covered by Puterman (2014, Chapter 8).

### 4.1. Limiting Average State-Action Frequencies

For every timestep $t$, $\mathrm{Pr}^{\pi, \mu}(S_t = s, A_t = a) \in [0, 1]$ is the probability that an agent will observe $(s, a)$ at time $t$ if it starts from $\mu$ and selects actions according to $\pi \in \Pi^{HR}$. A *limiting average state-action frequency* is an average of these probabilities taken over all timesteps,

$$\lim_{T \to \infty} \frac{1}{T} \sum_{t=0}^{T-1} \mathrm{Pr}^{\pi, \mu}(S_t = s, A_t = a). \qquad (1)$$

This limit does not always exist (see examples in Chapter 8 of Puterman (2014)'s book). So we need additional conditions to maximize objectives defined in terms of these limits.

### 4.2. Conditions on Transition Probabilities

For each stationary policy $\pi \in \Pi^{SR}$ given by decision rule $d^\pi$, let $P_\pi$ be the state-transition matrix with entry $[P_\pi]_{s',s}$ equal to $\sum_a d^\pi(a|s)P(s'|s,a)$.

**Definition 4.1** ((Puterman, 2014))**.** A transition probability function $P : \mathcal{S} \times \mathcal{A} \to \mathrm{Dist}(\mathcal{S})$ is said to be **unichain** if, for every stationary deterministic policy $\pi \in \Pi^{SD}$, the state-transition matrix $P_\pi$ has a single recurrence class plus a possibly empty set of transient states. A unichain transition probability function is said to be **aperiodic** if every recurrence class corresponding to a stationary deterministic policy $\pi \in \Pi^{SD}$ is aperiodic.

The unichain and aperiodic conditions are standard in average-reward MDP analysis, and they are discussed further in Appendix E. The unichain condition ensures that all stationary policies have well-defined occupancy measures. The aperiodic condition ensures that average state-action frequencies converge to their limits sufficiently fast.

## 4.3. Occupancy Measures

Following Puterman (2014, Section 8.9.1) we let $\Pi^1(\mu) \subseteq \Pi^{HR}$ be the set of all history-dependent policies whose limiting average state-action frequencies exist:

$$\Pi^1(\mu) = \{\pi : \text{ for all } s \text{ and } a, \text{the limit in (1) exists}\}. \quad (2)$$

**Definition 4.2.** For every policy $\pi \in \Pi^1(\mu)$, the policy's **occupancy measure** $x^{\pi,\mu} : \mathcal{S} \times \mathcal{A} \to [0,1]$ maps each state-action pair to its limiting average state-action frequency:

$$x^{\pi,\mu}(s,a) = \lim_{T \to \infty} \frac{1}{T} \sum_{t=0}^{T-1} \text{Pr}^{\pi,\mu}(S_t = s, A_t = a).$$

Every occupancy measure is a valid probability distribution over the set of state-action pairs. The sets of occupancy measures corresponding to HR, SR, and SD policies are defined as follows:

$$X^1(\mu) = \{x^{\pi,\mu} : \pi \in \Pi^1(\mu)\}$$
$$X^{SR}(\mu) = \{x^{\pi,\mu} : \pi \in \Pi^1(\mu) \cap \Pi^{SR}\}$$
$$X^{SD}(\mu) = \{x^{\pi,\mu} : \pi \in \Pi^1(\mu) \cap \Pi^{SD}\}$$

The set $X \subseteq \text{Dist}(\mathcal{S} \times \mathcal{A})$ contains all state-action distributions $x$ that satisfy,

$$\forall s' \in \mathcal{S}, \quad \sum_{a'} x(s',a') = \sum_{s,a} P(s'|s,a)x(s,a). \quad (3)$$

**Lemma 4.3.** *When the transition probability function is unichain:*

1. $\Pi^{SR} \subset \Pi^1(\mu)$.

2. $X^1(\mu) = X^{SR}(\mu) = X$.

3. *$X$ is equal to the convex hull of $X^{SD}(\mu)$. In particular, $X$ is closed and convex.*

4. *For every state-action distribution $x \in X$, the stationary policy $\pi \in \Pi^{SR}$ given by the decision rule $d^\pi$, where*

$$d^\pi(a|s) = \begin{cases} \frac{x(s,a)}{\sum_{a'} x(s,a')} & \text{if } \sum_{a'} x(s,a') > 0 \\ \text{arbitrary} & \text{otherwise,} \end{cases}$$

*satisfies $x^{\pi,\mu} = x$.*

*Proof.* See Appendix C.1. $\qquad\square$

Part (2) shows that, when the transition probability function is unichain, occupancy measures do not depend on $\mu$. So, under the unichain condition, we will write $\Pi^1$ instead of $\Pi^1(\mu)$ and, for each $\pi \in \Pi^1(\mu)$, write $x^\pi$ instead of $x^{\pi,\mu}$.

## 4.4. Markov Decision Processes

**Definition 4.4.** A **finite Markov decision process** $(\mathcal{S}, \mathcal{A}, P, \mu, r)$ consists of a finite environment $(\mathcal{S}, \mathcal{A}, P, \mu)$ and a reward function $r : \mathcal{S} \times \mathcal{A} \to \mathbb{R}$. A finite Markov decision process is **unichain** if its transition probability function is unichain, and is **aperiodic** it its transition probability function is aperiodic.

For our purposes, a policy $\pi \in \Pi^1(\mu)$ is *optimal under the average-reward criterion* if

$$\sum_{s,a} x^{\pi,\mu}(s,a)r(s,a) = \sup_{\pi' \in \Pi^1(\mu)} \sum_{s,a} x^{\pi',\mu}(s,a)r(s,a). \quad (4)$$

Within finite unichain Markov decision processes, there always exists a deterministic stationary policy that is optimal under this criterion (Puterman, 2014, Theorem 8.8.6).

## 5. Markov Decision Contests

**Definition 5.1.** A **finite Markov decision contest** $(\mathcal{S}, \mathcal{A}, P, \mu, M)$ consists of a finite environment $(\mathcal{S}, \mathcal{A}, P, \mu)$ and a Markov preference margin $M : (\mathcal{S} \times \mathcal{A}) \times (\mathcal{S} \times \mathcal{A}) \to \mathbb{R}$. A Markov decision contest is **unichain** if its transition probability function is unichain, and it is **aperiodic** if its transition probability function is aperiodic.

Recall from (2) that $\Pi^1(\mu)$ is the set of all history-dependent policies whose limiting state-action frequencies exist.

**Definition 5.2.** Within a finite Markov decision contest, a policy $\pi \in \Pi^1(\mu)$ is **optimal under the average-preference-margin criterion** if, for every other policy $\pi' \in \Pi^1(\mu)$,

$$\sum_{s,a} \sum_{s',a'} x^{\pi,\mu}(s,a)x^{\pi',\mu}(s',a')M((s,a),(s',a')) \geq 0.$$

Policies that are optimal under this criterion will be called *solutions* to the finite Markov decision contest.

### 5.1. Examples

**Example 5.3** (Markov decision processes)**.** Every Markov decision process $(\mathcal{S}, \mathcal{A}, P, \mu, r)$ can be represented as a Markov decision contest $(\mathcal{S}, \mathcal{A}, P, \mu, M_r)$, where $M_r : (\mathcal{S} \times \mathcal{A}) \times (\mathcal{S} \times \mathcal{A}) \to \mathbb{R}$ is given by

$$M_r((s,a),(s',a')) = r(s,a) - r(s',a'). \quad (5)$$

A policy is optimal under the average-preference-margin criterion within $(\mathcal{S}, \mathcal{A}, P, \mu, M_r)$ if and only if it is optimal under the average-reward criterion within $(\mathcal{S}, \mathcal{A}, P, \mu, r)$.

Conversely, a preference margin $M$ can be expressed in the form of (5) only if it satisfies, for all $(s, a), (s', a'), (s'', a'')$,

$$M((s, a), (s'', a''))$$
$$= M((s, a), (s', a')) + M((s', a'), (s'', a'')).$$

In particular, this form does not hold for *nontransitive* preference margins, where there are state-action pairs for which $M((s, a), (s', a')) > 0$, $M((s', a'), (s'', a'')) > 0$, and $M((s'', a''), (s, a)) > 0$.

**Example 5.4** (Best two-out-of-three rule). May (1954) observed that preferences are often nontransitive when based on multiple features. For instance, if each state-action pair is evaluated according to three features, (say position, speed, and stability of a robot), then preferring the pair that is better on two of the three features leads to a nontransitive preference margin.

**Example 5.5** (Aggregated preferences). Even when an individual's preferences are transitive, nontransitive preferences can arise when aggregating preferences from multiple individuals. Consider, for instance, the preference margin $M_{\mathrm{agg}}$ representing the fraction of individuals preferring $(s, a)$ to $(s', a')$ minus the fraction of individuals preferring $(s', a')$ to $(s, a)$. So, $M_{\mathrm{agg}}((s, a), (s', a')) \geq 0$ if and only if at least half of the individuals prefer $(s, a)$ to $(s', a')$. It is well known that $M_{\mathrm{agg}}$ can be nontransitive (Fishburn, 1984; Gehrlein, 1983). This is known as Condorcet's paradox.

### 5.2. Existence of Optimal Stationary Policies

Lemma 4.3 showed that the set of all occupancy measures is equal to the set of occupancy measures that correspond to randomized stationary policies. This leads to our first important result about solving Markov decision contests.

**Theorem 5.6.** *Within all finite unichain Markov decision contests:*

1. *There exists a randomized stationary policy that is optimal under the average-preference-margin criterion.*

2. *A randomized stationary policy is optimal under the average-preference-margin criterion if, and only if, it is a solution to*

$$\max_{\pi \in \Pi^{SR}} \min_{\pi' \in \Pi^{SR}} \sum_{s,a} \sum_{s',a'} x^\pi(s, a) x^{\pi'}(s', a') M((s, a), (s', a')).$$
(6)

*Proof.* See Appendix C.2. ☐

This theorem simplifies the problem of solving Markov decision contests in two ways. First, part (1) guarantees that optimal stationary randomized policies exist. This means an agent can behave optimally without needing to store or implement a history-dependent decision rule. This is analogous to the result guaranteeing existence of optimal stationary deterministic policies for average-reward MDPs (Puterman, 2014, Theorem 8.8.6).

Second, part (2), it shows that, to solve these contests, it suffices to solve a two-player, zero-sum game between stationary randomized policies. This avoids the need to store and compute history-dependent policies. Thus, any solution method for finite unichain Markov decision contests can safely restrict itself to stationary policies.

## 6. Exact Solution Methods

In finite unichain Markov decision contests, an optimal policy can be recovered exactly in time polynomial in $|\mathcal{S}||\mathcal{A}|$ (Theorem 6.1).

This result does not follow from standard game-solving methods. The standard linear program (LP) for solving the game in (6) would run in time polynomial in $|\mathcal{A}|^{|\mathcal{S}|}$. This is because the game is played over the set of occupancy measures, which is the convex hull of occupancy measures of stationary deterministic policies (Lemma 4.3). There are $|\mathcal{A}|^{|\mathcal{S}|}$ stationary deterministic policies, which makes the standard LP for solving (6) polynomial in $|\mathcal{A}|^{|\mathcal{S}|}$.

But, when the Markov decision contest is unichain, the set of occupancy measures can be represented as the solutions to the system of $|\mathcal{S}|$ linear equations in (3). This reduces the cost of solving Markov decision contests exactly.

**Theorem 6.1.** *For every finite unichain Markov decision contest, there exists a linear program that solves the game in (6) using $|\mathcal{S}||\mathcal{A}| + |\mathcal{S}| + 1$ variables and $2|\mathcal{S}||\mathcal{A}| + |\mathcal{S}| + 1$ constraints.*

*Proof.* See Appendix C.3. ☐

Consequently, every finite unichain Markov decision contest is solvable in time polynomial in $|\mathcal{S}||\mathcal{A}|$ (instead of $|\mathcal{A}|^{|\mathcal{S}|}$).

In particular, this result shows that Markov decision contests (with the average-preference-margin criterion) are in the same complexity class as Markov decision processes (with the average-reward criterion). This is surprising, given that Markov decision contests are a strict generalization of Markov decision processes (refer to 5.1).

Exact solution methods are not suitable for large environments, as the linear dependence on $|\mathcal{S}|$ is problematic. We turn to approximate solution methods next, to address these cases.

# 7. Approximate Solution Methods

We now present a simple iterative algorithm that converges to an optimal policy at a sublinear rate (Theorem 7.5). The definitions and results of this section are all defined with respect to a finite Markov decision contest that is assumed to be unichain.

As is common in two-player, zero-sum games, convergence is described in terms of the optimality gap.

**Definition 7.1.** The **optimality gap** of a stationary policy $\pi \in \Pi^{SR}$ is

$$\max_{\pi' \in \Pi^{SR}} \sum_{s,a} \sum_{s',a'} x^{\pi'}(s,a) x^{\pi}(s',a') M((s,a),(s',a')).$$
(7)

The optimality gap is nonnegative, and it is equal to zero if and only if $\pi$ is an optimal policy. We will also define $\bar{M}(\pi, \pi)$ as the objective being maximized in (7):

$$\bar{M}(\pi, \pi') = \sum_{s,a} \sum_{s',a'} x^{\pi}(s,a) x^{\pi'}(s',a') M((s,a),(s',a')).$$

## 7.1. Challenges with Standard Approximate Methods

In principle, it is possible to solve the game in (6) with standard methods such as online mirror descent or fictitious play. By Lemma 4.3, the game in (6) is played over the set of occupancy measures, which is equal to the convex hull of the set of occupancy measures corresponding to stationary deterministic policies. Thus, one can solve this game by maintaining a probability distribution over the set of stationary deterministic policies, and updating the probability distribution iteratively.

But, as we discuss in Appendix F, updating this distribution requires knowing the exact probability of choosing each stationary deterministic policy. This seems difficult to obtain or estimate when decision rules are represented with function approximators.

Instead, we propose a new learning algorithm which is more amenable to function approximation. It makes use of two new functions to measure policy performance. We introduce these functions next.

## 7.2. Marginal values

The marginal value function measures how much utility is gained or lost by starting in state $s$ and following $\pi$, compared to starting from a state sampled from $\pi$'s steady-state distribution.

The *preference-margin cumulant* $c_M^{\pi} : \mathcal{S} \times \mathcal{A} \to [-M_{\max}, M_{\max}]$ measures the expected utility gained (or lost) from observing a state-action pair $(s, a)$ instead of a

state-action pair sampled from the occupancy measure of $\pi$:

$$c_M^{\pi}(s,a) = \sum_{s',a'} x^{\pi}(s',a') M((s,a),(s',a')).$$
(8)

When $c_M(s, a) > 0$, $(s, a)$ is preferable to a random sample from $x^{\pi}$. For every timestep $t$ and start state $s_0$,

$$E_t^{\pi}[c_M^{\pi}(S_t, A_t)|s_0 = s]$$

is the expected value of the cumulant at time $t$.

**Definition 7.2.** For every stationary policy $\pi \in \Pi^{SR}$ and state $s \in \mathcal{S}$, the **marginal value of $s$ under $\pi$**, denoted by $V_M^{\pi}(s)$, is the sum of all preference-margin cumulants expected under $\pi$ when starting in $s$:

$$V_M^{\pi}(s) = \sum_{t=0}^{\infty} E_t^{\pi}[c_M^{\pi}(S_t, A_t)|S_0 = s].$$
(9)

The **marginal state-action value of $(s, a)$ under $\pi$**, denoted by $Q_M^{\pi}(s, a)$, is the sum of all preference margin cumulants expected when starting in $s$, taking action $a$, and following $\pi$ thereafter:

$$Q_M^{\pi}(s,a) = \sum_{t=0}^{\infty} E_t^{\pi}[c_M^{\pi}(S_t, A_t)|S_0 = s, A_0 = a].$$
(10)

When the Markov decision contest is periodic, marginal values and state-action values may diverge (for the same reason that differential values and state-action values diverge when the MDP is periodic). When the contest is aperiodic, they have the following basic properties.

**Lemma 7.3** (Properties of marginal value functions). *If the Markov decision contest is aperiodic (in addition to being unichain), then there exists a constant $\tau$ such that, for every stationary policy $\pi \in \Pi^{SR}$, state $s \in \mathcal{S}$, and action $a \in \mathcal{A}$,*

*1. $|V_M^{\pi}(s)| \leq M_{max}\tau$ and $|Q_M^{\pi}(s, a)| \leq 2 M_{max} \tau$.*

*2. $Q_M^{\pi}(s, a) = c_M^{\pi}(s, a) + \sum_{s'} P(s' \mid s, a) V_M^{\pi}(s')$.*

*3. $V_M^{\pi}(s) = \sum_{a'} d^{\pi}(a' \mid s) Q_M^{\pi}(s, a')$, where $d^{\pi}$ is the Markov decision rule that determines $\pi$.*

*Proof.* See Appendix C.4. □

Marginal values are useful, as they allow us to represent long-term performance in terms of the expected difference between marginal values and state-action values.

**Lemma 7.4** (Performance Difference Lemma). *If the Markov decision contest is aperiodic (in addition to being unichain), then, for all stationary policies $\pi, \pi' \in \Pi^{SR}$,*

$$\bar{M}(\pi, \pi') = \sum_{s,a} x^{\pi}(s,a)(Q_M^{\pi'}(s,a) - V_M^{\pi'}(s)).$$

---

**Algorithm 1** Hedged Policy Iteration (Tabular Form)

1: **Input:** learning rate $\eta$
2: Initialize decision rule $d_1$ as $d_1(a \mid s) = 1/|\mathcal{A}|$
3: Initialize $\bar{x}_0$ as $\bar{x}_0(s,a) = 0$
4: **for** $k = 1$ **to** $K$ **do**
5:    #1. Compute occupancy measure
6:    Compute occupancy measure $x^{\pi_k}$
7:    $\bar{x}_k \leftarrow \bar{x}_{k-1} + (x^{\pi_k} - \bar{x}_{k-1})/k$
8:    # 2. Evaluate
9:    Compute $Q_M^{\pi_k}(s,a)$
10:   # 3. Update decision rule
11:   Set $d_{k+1}(a|s) \propto d_k(a|s) \exp(\eta\, Q_M^{\pi_k}(s,a))$
12: **end for**
13: # 4. Compute decision rule of the return policy
14: $d_K^{\text{HPI}}(a|s) \leftarrow \bar{x}_K(s,a) / \sum_{a'} \bar{x}_K(s,a')$
15: **Return** $d_K^{\text{HPI}}$

---

*Proof.* See Appendix C.5. □

In particular, this lemma implies that, for all policies $\pi, \pi' \in \Pi^{SR}$, if the decision rule $d^\pi$ that determines $\pi$ satisfies $\sum_a d^\pi(a|s) Q_M^{\pi'}(s,a) \geq V_M^{\pi'}(s)$ for all $s$, then

$$\bar{M}(\pi, \pi') \geq 0.$$

### 7.3. Hedged Policy Iteration

Algorithm 1 approximately solves Markov decision contests. It iteratively produces randomized stationary policies $\pi_1, \pi_2, \ldots$ that are given by Markov decision rules $d_1, d_2, \ldots$. Decision rules are chosen with the Hedge algorithm (Freund & Schapire, 1995). And so, our algorithm is called *Hedged Policy Iteration* (HPI).

Because the Hedge algorithm does not converge in last-iterate, neither does HPI. Instead, HPI returns a decision rule $d_K^{\text{HPI}}$ such that the occupancy measure of the policy $(d_K^{\text{HPI}})_{t=0}^\infty$ is equal to the average occupancy measure $\bar{x}_K$. The policy $(d_K^{\text{HPI}})_{t=0}^\infty$ is called the *return policy of Hedged Policy Iteration*. It has the following convergence guarantee.

**Theorem 7.5.** *Suppose that the Markov decision contest aperiodic (in addition to being unichain). When Hedged Policy Iteration (Algorithm 1) runs for $K \geq \log(|\mathcal{A}|)$ iterations and its learning rate $\eta$ is equal to $(2\, M_{max}\, \tau)^{-1} \sqrt{\log(|\mathcal{A}|)/K}$, the optimality gap its return policy is no greater than*

$$4\, M_{max}\, \tau \sqrt{\frac{\log(|\mathcal{A}|)}{K}}.$$

*Proof.* See Appendix C.6. □

HPI's rate of convergence does not depend explicitly on $|\mathcal{S}|$. The reason is that HPI updates the action probabilities of

---

**Algorithm 2** HPI (Policy-Gradient Form)

1: **Input:** learning rate $\eta$
2: Initialize decision rule $d_1$ as $d_1(a \mid s) = 1/|\mathcal{A}|$
3: Initialize $\bar{x}_0$ as $\bar{x}_0(s,a) = 0$
4: **for** $k = 1$ **to** $K$ **do**
5:    #1. Compute occupancy measure
6:    Compute occupancy measure $x^{\pi_k}$
7:    $\bar{x}_k \leftarrow \bar{x}_{k-1} + (x^{\pi_k} - \bar{x}_{k-1})/k$
8:    # 2. Evaluate
9:    Compute $Q_M^{\pi_k}(s,a)$
10:   # 3. Update decision rule
11:   Select $d_{k+1}$ by solving:
12:   $\max_\theta \sum_{s,a} x^{\pi_k}(s,a) \frac{d_\theta(a|s)}{d_k(a|s)} \left( Q_M^{\pi_k}(s,a) - \frac{1}{\eta} \log \frac{d_\theta(a|s)}{d_k(a|s)} \right)$
13: **end for**
14: # 4. Compute decision rule of the return policy
15: Compute $d_K^{\text{HPI}}$ by solving:
16: $\max_\theta \sum_{s,a} \bar{x}_K(s,a) \log d_\theta(a|s)$
17: **Return** $d_K^{\text{HPI}}$

---

each state in parallel. However, the rate depends implicitly on $|\mathcal{S}|$ through $\tau$: if $|\mathcal{S}|$ is large, then it may take policies longer to converge to their occupancy measures. This can make $\tau$ larger.

### 7.4. A Policy-Gradient Form

When decision rules are parameterized with parameter vectors, the decision-rule updates in steps #3 and #4 of Algorithm 1 are no longer practical. Fortunately, both updates can be re-expressed as optimization problems that are convex in the decision rule's action probabilities. Parameterized decision rules can be updated by applying stochastic gradient ascent to these optimization problems.

Algorithm 2 presents HPI with the decision rule update steps in their alternate forms. We call this the *policy-gradient (PG) form* of HPI. The new form for step #3 is made possible by the properties of the softmax update rule. The new form of step #4 is justified with the following lemma.

**Lemma 7.6.** *Let $x \in X$ be an occupancy measure and $\pi \in \Pi^{SR}$ be a stationary policy given by decision rule $d^\pi$. Then, the occupancy measure of $\pi$ is equal to $x$ if and only if $d^\pi$ is a solution to*

$$\max_\theta \sum_{s,a} x(s,a) \log d_\theta(a|s).$$

*Proof.* See Appendix C.7. □

This proof has several variants in the behaviour-cloning literature (e.g. (Syed et al., 2008)). We use it here to show the following result.

**Algorithm 3** HPI (Deep Learning Implementation)

1: **Input**: learning rate $\eta$, buffer $\mathcal{B}_{\text{avg}}$, number of behavior cloning steps $n_{\text{BC}}$,
2: Initialize decision rule $\hat{d}_1 : \mathcal{S} \to \text{Dist}(\mathcal{A})$ arbitrarily
3: Initialize value estimate $\hat{Q}_M^{\pi_1} : \mathcal{S} \times \mathcal{A} \to \mathbb{R}$ arbitrarily
4: **for** $k = 1, \dots, K$ **do**
5:     # 1. Estimate occupancy measure
6:     Collect samples $\mathcal{B} = \{(s_t^i, a_t^i)_{t=0}^{T-1}\}_{i=1}^N$ from $\pi_k$
7:     $\mathcal{B}_{\text{avg}} \leftarrow \text{UpdateBuffer}(\mathcal{B}_{\text{avg}}, \mathcal{B})$
8:     # 2. Evaluate (approximately)
9:     $\hat{c}_{k,t}^i \leftarrow \frac{1}{|\mathcal{B}|} \sum_{s',a' \in \mathcal{B}} M((s_t^i, a_t^i), (s', a'))$
10:    $\hat{Q}_M^{\pi_k} \leftarrow \text{UpdateQ}(\hat{Q}_M^{\pi_{k-1}}, \{(s_t^i, a_t^i, \hat{c}_{k,t}^i)_{t=0}^{T-1}\}_{i=1}^N)$
11:    # 3. Update decision rule
12:    Select $\hat{d}_{k+1}$ by optimizing:
13:    $\max_\theta \frac{1}{NT} \sum_{s_t^i, a_t^i} \frac{d_\theta(a_t^i|s_t^i)}{d_k(a_t^i|s_t^i)} \left( \hat{Q}_M^{\pi_k}(s_t^i, a_t^i) - \frac{1}{\eta} \log \frac{d_\theta(a_t^i|s_t^i)}{d_k(a_t^i|s_t^i)} \right)$
14: **end for**
15: # 4. Estimate average policy
16: Compute $\hat{d}_K^{\text{HPI}}$ by optimizing, for $n_{\text{BC}}$ gradient steps,
17: $\max_\theta \frac{1}{|\mathcal{B}_{\text{avg}}|} \sum_{s,a \in \mathcal{B}_{\text{avg}}} \log d_\theta(a|s)$
18: **Return** $\hat{d}_K^{\text{HPI}}$

**Proposition 7.7.** *If the Markov decision contest aperiodic (in addition to being unichain), then Algorithm 2 converges at the rate described in Theorem 7.5.*

*Proof.* See Appendix C.8. □

Thus, the policy-gradient form of Hedged Policy Iteration is amenable to function approximation; its maximization objectives are convex with respect to the action probabilities of the decision rule $d_\theta$. Next, we discuss how its remaining quantities can be approximated using standard methods from deep reinforcement learning.

### 7.5. A Deep-Learning Implementation

From the policy gradient form of HPI, the deep learning implementation of HPI (Algorithm 3) is fairly straightforward. To obtain Algorithm 3 from Algorithm 2:

- Replace the occupancy measure $x_k$ with a buffer of samples $\mathcal{B}$ collected from $\pi_k$. Replace the average occupancy measure $\bar{x}_k$ with a buffer of samples $\mathcal{B}_{\text{avg}}$.

- Replace exact computation of $Q_M^{\pi_k}(s, a)$ with a value function update, UpdateQ. Because the cumulant has the same type as a reward function, any deep reinforcement learning method for value estimation can be used for this update.

- Calculate the decision rules $\hat{d}_{k+1}$ and $\hat{d}_K^{\text{HPI}}$ by solving the optimization objectives in steps #3 and #4 approx-

imately. By default, when $n_{BC} = 0$, the last iterate $\hat{d}_{k+1}$ is returned.

**HPI-Clip.** As we discuss in Appendix H.1, the approximate update step in step #3 of Algorithm 3 can be replaced by a clipped surrogate objective, similar to the clipped objective in Proximal Policy Optimization (Schulman et al., 2017). When this replacement is made, we refer to the resulting algorithm as *HPI-Clip*.

## 8. Experiments

One of our main reasons for introducing Markov decision contests was our claim that existing methods of reinforcement learning with pairwise preferences were inefficient in long-term decision problems. We now validate this claim, through the following hypothesis:

*In long-term decision problems, HPI is more learning-efficient than existing algorithms that learn with pairwise preferences.*

**Method.** We compared HPI (Algorithm 3) and HPI-Clip (Algorithm 4) against Swamy et al. (2024)'s Self-Play Preference Optimization (SPPO). We did not compare with the algorithms of Shani et al. (2024) and Wu et al. (2026), as they both solve a regularized objective. Appendix H provides more details about our algorithm implementations.

We considered 13 Markov decision contests (called "tasks", for short), which used Todorov et al. (2012)'s Mujoco environments. The horizon for each task is long: the default horizon is 2048 steps (though some tasks end early, when specific states are reached).

*Mujoco-v5 Suite.* The first 11 tasks were the eleven tasks of the Mujoco-v5 suite (Todorov et al., 2012). This is a common set of continuous control tasks, which have large state and action spaces. For these tasks, the goal is specified with a reward function. So the preference margin given to HPI, HPI-Clip, and SPPO was given as the difference in task reward (as in (5)). We also compared with PPO, which learns from task's reward directly. The optimality gap in these tasks is equal to the episodic return, up to constant factors. And so, we measured learning efficiency as the Area Under the Curve (AUC) of the episodic return curve over the environment timestep.

*Mujoco-NT Tasks.* The last two tasks used the environments from the Mujoco-v5 suite (Reacher and Walker2d), but replaced the task's original reward with a nontransitive preference margin. We call these the *Mujoco-NT Tasks*. The nontransitive preference margins are described in Appendix A.2. The Reacher-NT preference margin was proposed by Swamy et al. (2024). The Walker2d-NT preference margin was defined with a best-two-out-of three rule (as in

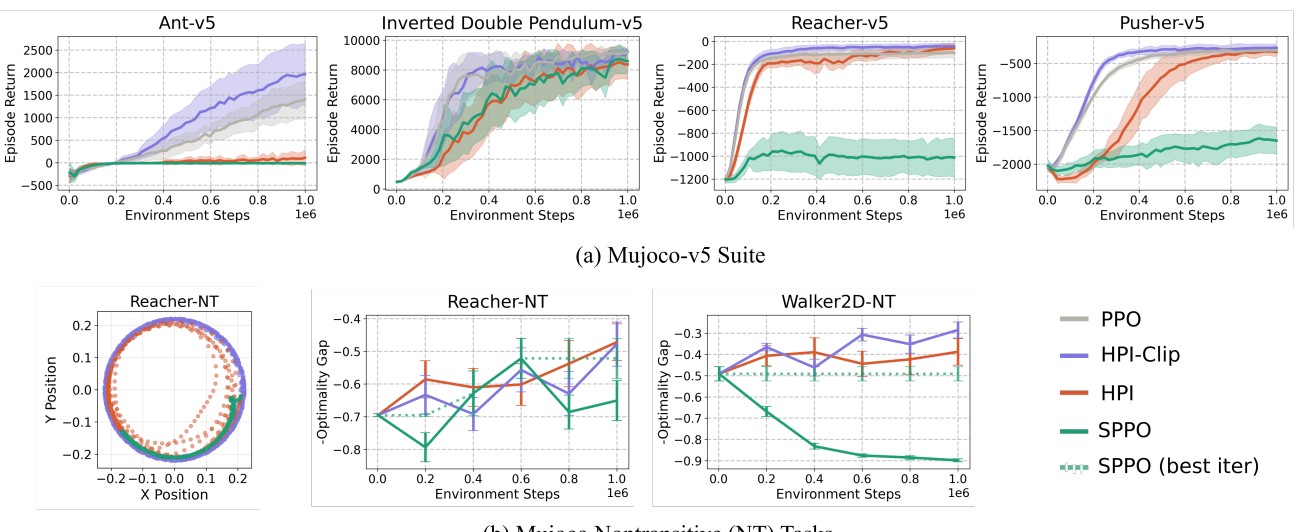

(a) Mujoco-v5 Suite

(b) Mujoco Nontransitive (NT) Tasks

*Figure 1.* Performance of algorithms on (a) Mujoco-v5 Suite and (b) Mujoco-NT Tasks, which have long time horizons. Both HPI-Clip and HPI are more learning-efficient than SPPO on these tasks, as measured by the Area Under the Curve (AUC).

Example 5.4), with performance criteria on three features. In Reacher-NT, the optimal policy must trace a circle with the reacher fingertip. In Walker2d-NT, the optimal policy must visit each of the three features with equal probability.

We estimated the optimality gap of these policies by training a PPO algorithm to minimize the objective in (7). We also provide trajectory plots to ensure that the estimated optimality gap was consistent with expected behavior. Learning efficiency was measured as the Area Under the Curve of the (inverted) optimality gap curve over the environment timestep. Additional evaluation details are provided in Appendix A.2.

**Results.** All reported results show the mean and 95% confidence intervals across 10 independent instances with different random seeds. Figure 1 shows a summary of our results. The remaining results on the Mujoco-v5 suite are shown in Figure 2. Figures 3 and 4 show the remaining results for the Mujoco-NT tasks. **In 12/13 tasks, HPI's AUC confidence interval either exceeds or overlaps with SPPO's.** The same holds for HPI-Clip's AUC confidence interval in comparison with SPPO's. Thus, we conclude that both HPI and HPI-Clip are significantly more learning-efficient than SPPO on this set of tasks.

**Ablation Studies.** In Appendix A.3 we verify that similar gains in learning efficiency hold when algorithms attempt to solve the tasks using preference models that are learned from offline data. In Appendix A.4, we provide a detailed comparison of HPI-Clip and SPPO to study how learning efficiency varies with the task horizon.

## 9. Conclusion

Existing problem models for reinforcement learning with pairwise preferences assume a finite time horizon, and they lack guarantees on the performance of Markov policies relative to history-dependent ones. We introduced Markov decision contests to address these limitations, and proved three results. First, *stationary* policies are optimal within the set of history-dependent policies (Theorem 5.6). Second, solving a Markov decision contest exactly is in P, the same complexity class as Markov decision processes, despite the greater generality of pairwise preferences (Theorem 6.1). Third, a simple iterative algorithm (Hedged Policy Iteration) solves Markov decision contests approximately, converging at a rate of $1/\sqrt{K}$ (Theorem 7.5). In 13 high-dimensional Markov decision contests with long time horizons, our algorithms (HPI and HPI-Clip) were significantly more learning-efficient than SPPO, the most relevant prior algorithm.

These results are relevant to applications of reinforcement learning with pairwise preferences for large language models that attempt tasks with long time horizons. But more investigation is needed. It remains unclear when the gains from learning with preference margins outweigh those from learning with reward functions, and how effective these methods are at aligning language models in practice. Answering these questions would bring the theory and practice of reinforcement learning with pairwise preferences considerably closer together.

## Acknowledgements

The reviewers provided excellent feedback on this paper, which greatly improved it. Wesley Chung, Karim Abdel

Sadek, Henrik Marklund, David Abel, Mandana Samiei, Shahrad Mohammadzadeh, and Anmol Kagrecha gave ideas and suggestions that were very helpful.

This research was enabled in part by compute resources provided by Mila (mila.quebec). It was funded in part by a Mitacs Globalink Research Award.

## Impact Statement

This paper presents work whose goal is to advance the field of Machine Learning. There are many potential societal consequences of our work, none which we feel must be specifically highlighted here.

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

# A. Additional Experiments

## A.1. Mujoco-v5 Suite

Figure 2 illustrates the performance of our algorithms in the Mujoco-v5 suite (Todorov et al., 2012). The solid lines represent means over 10 independent training runs and the shaded areas represent 95% confidence intervals.

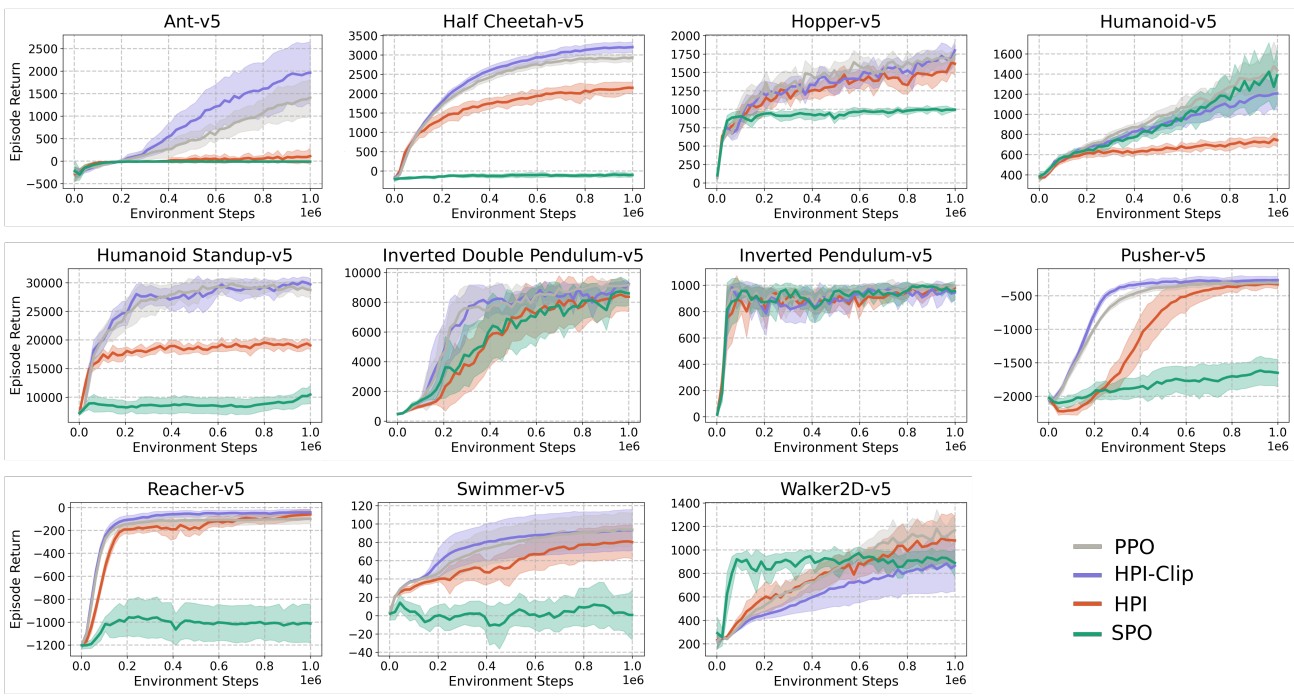

*Figure 2.* Comparison of algorithms on the Mujoco-v5 suite, training for one million timesteps.

We measured learning efficiency as the Area Under the Curve (AUC) over all training steps. The confidence interval for HPI-Clip's AUC was strictly higher than the confidence interval for SPPO's AUC in 7 tasks (Ant-v5, Half Cheetah-v5, Hopper-v5, Humanoid Standup-v5, Pusher-v5, Reacher-v5, Swimmer-v5). Meanwhile, the confidence interval for SPPO's AUC was strictly higher than the confidence interval for HPI-Clip's AUC just 1 task (Walker2d-v5). We conclude that HPI-Clip is more learning-efficient than SPPO on the Mujoco-v5 suite, because its AUC is significantly better on 7 of 11 tasks and only significantly worse on 1 of 11 tasks.

The confidence interval for HPI's AUC was strictly higher than the confidence interval for SPPO's AUC in 6 tasks (Half Cheetah-v5, Hopper-v5, Humanoid Standup-v5, Pusher-v5, Reacher-v5, Swimmer-v5). SPPO's AUC confidence interval was strictly higher than HPI-Clip's AUC confidence interval on 1 task (Humanoid-v5). So, we conclude that HPI is more learning efficient than SPPO on the Mujoco-v5 suite, because its AUC is significantly better on 6 of 11 tasks and only statistically worse on 1 of 11 tasks.

## A.2. Mujoco-NT

The Mujoco-NT tasks were adapted from Reacher-v5 and Walker2d-v5 tasks from the Mujoco suite. These environments were chosen as they were relatively fast to run (which was necessary for our evaluation, as we will discuss).

**Task description.** The Mujoco-NT tasks, Reacher-NT and Walker2d-NT, kept all of the same elements of the Reacher-v5 and Walker2d-v5 tasks, except they replaced the task rewards from the Mujoco suite with preference margins.

The Reacher-NT preference margin has two preference criteria: fingertip radius from origin and first arm angular position. The margin favors observations where the fingertip is farther from the origin (30% weight) and implements a counterclockwise angular preference for arm 1 (70% weight). Swamy et al. (2024) considered a nontransitive preference of this

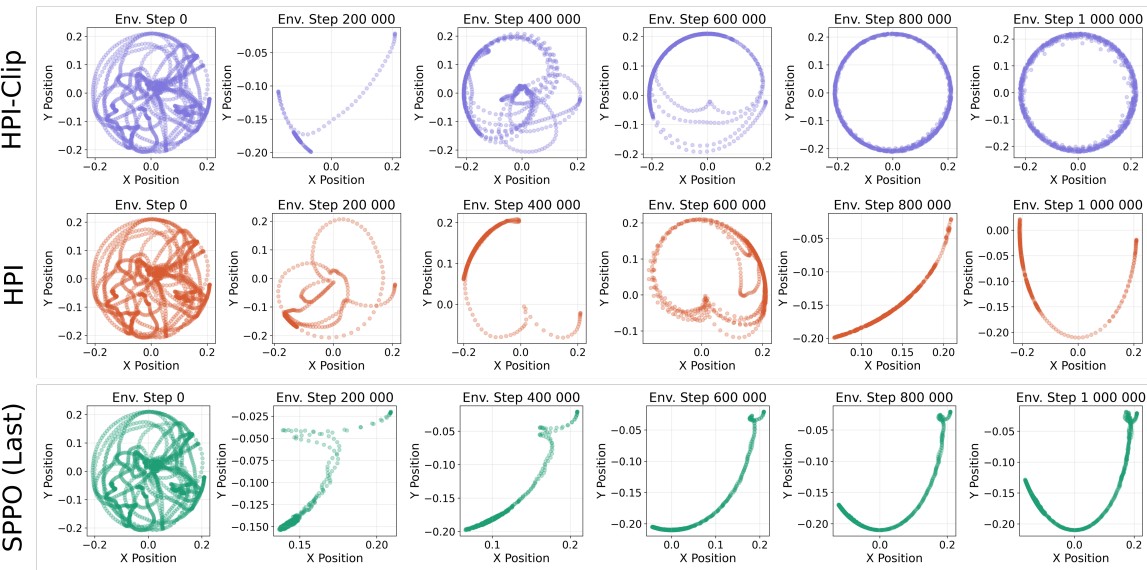

*Figure 3.* Position of the Reacher's fingertip in the Reacher-NT task during training. In this task, the optimal policy creates a perfect circle with the fingertip.

form for the Ant environment. We chose to experiment with Reacher instead of Ant, because we found the optimality gap estimates to be more reliable in Reacher. The pseudocode for this preference margin is provided in Appendix H.4.

The Walker2d-NT preference margin implements a nontransitive preference based on three competing objectives: Height > Speed > Stability > Height. Each environment observation is classified by its dominant feature (according to its highest normalized value), and preferences follow a rock-paper-scissors pattern where high walkers beat fast walkers, fast walkers beat stable walkers, and stable walkers beat high walkers. The pseudocode for this preference margin is given in Appendix H.4.

**Evaluation details.**    To evaluate performance in Mujoco-NT, we estimated the optimality gap of each algorithm (HPI, HPI-Clip, and SPPO) after every 200 000 timesteps. After each 200 000 timesteps, we would estimate the occupancy measure the algorithm's return policy by sampling 100 trajectories from it. Then, we trained a PPO agent the optimality gap objective from (7). The PPO agents used the same hyperparameters and network architectures as in the Mujoco-v5 suite.

Because this was only an approximation of the true optimality gap, we also plotted trajectories sampled from each algorithm throughout training, to determine whether the trends in approximate optimality gap were consistent with policy behavior.

**Results.**    The estimated optimality gaps are plotted in Figure 1(b). They are inverted so that the trends are visually similar to the expected return learning curves. The trajectory samples of algorithms run on Reacher-NT and Walker2d-NT are shown in Figures 3 and 4, respectively.

**Conclusions.**    We measure learning efficiency according to the AUC of the (inverted) optimality-gap learning curves in Figure 1(b). The AUC confidence intervals of HPI-Clip are strictly higher than the confidence intervals of SPPO on both Reacher-NT and Walker2d-NT. The same holds for the confidence intervals of HPI. Thus, we conclude that both HPI and HPI-Clip are more learning efficient than SPPO on this task.

Importantly, the trends in optimality gap are consistent with the actual behavior of decision policies in these tasks. For example, as the optimality gap shrinks in Reacher-NT, the reacher's fingertip position makes a large circle in Figure 3. As the optimality gap shrinks in Walker2d-NT, the policies spend a more even amount of time across all three dominant features in Figure 4.

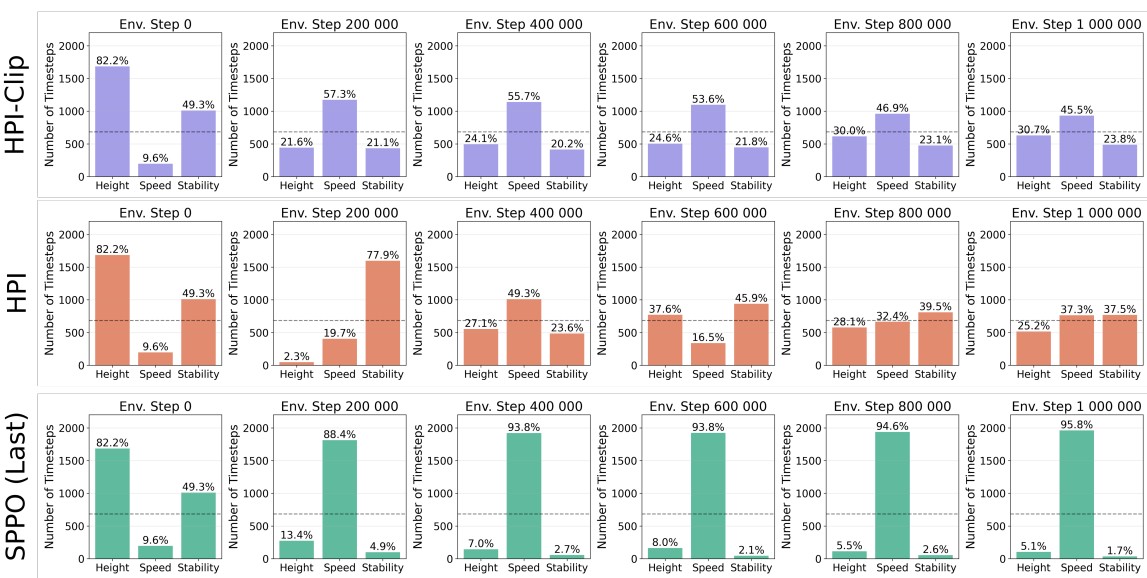

*Figure 4.* Dominant feature distribution in the Walker2d task during training. In this task, the optimal policy visits each dominant feature 33.3% of the time.

## A.3. Ablation 1: Experiments with a Learned Preference Margin

In many applications, the true preference margin is unknown, and it must be learned from pairwise comparison data. So, we conducted an experiment to verify whether the gains in learning efficiency we observed in Mujoco-v5 suite and Mujoco-NT would persist if the algorithms used a learned preference model.

For this experiment, we ran each algorithm in each task using a learned preference model. For each task and each algorithm, our method was the following:

1. Attempt the task using the tasks true preference margin (or the task's reward, for PPO). Save 500 trajectories that are evenly-spaced throughout the first 1 million environment steps, as it attempts the task using the task's preference margin (or the task's reward, for PPO).

2. Randomly sample 100,000 comparisons from the 500 rollouts.

3. Follow the procedure of Christiano et al. (2017) to learn an ensemble of three preference models (or an ensemble of three reward models, for PPO).

4. Attempt the task using the learned preference-model ensemble (or reward-model ensemble, for PPO).

### A.3.1. MUJOCO-V5 SUITE WITH LEARNED PREFERENCE MODELS

Figure 5 shows the performance of each algorithm that used a learned preference model to solve the Mujoco-v5 tasks.

Again, we measured learning efficiency as the Area Under the Curve (AUC) over all training steps. The confidence interval for HPI-Clip's AUC was strictly higher than the confidence interval for SPPO's AUC in 7 tasks (Ant-v5, Half Cheetah-v5, Hopper-v5, Humanoid-v5, Humanoid Standup-v5, Pusher-v5, Reacher-v5). Meanwhile, the confidence interval for SPPO's AUC was strictly higher than the confidence interval for HPI-Clip's AUC just 1 task (Inverted Pendulum-v5). We conclude that HPI-Clip is more learning-efficient than SPPO on the Mujoco-v5 suite, because its AUC is significantly better on 7 of 11 tasks and only significantly worse on 1 of 11 tasks.

The confidence interval for HPI's AUC was strictly higher than the confidence interval for SPPO's AUC in 8 tasks (Ant-v5, Half Cheetah-v5, Hopper-v5, Humanoid-v5, Humanoid Standup-v5, Inverted Double Pendulum-v5, Pusher-v5, Reacher-v5). SPPO's AUC confidence interval was strictly higher than HPI-Clip's AUC confidence interval on 1 task (Inverted Pendulum-v5). We conclude that HPI is more learning efficient than SPPO on the Mujoco-v5 suite, because its AUC is significantly better on 8 of 11 tasks and only statistically worse on 1 of 11 tasks.

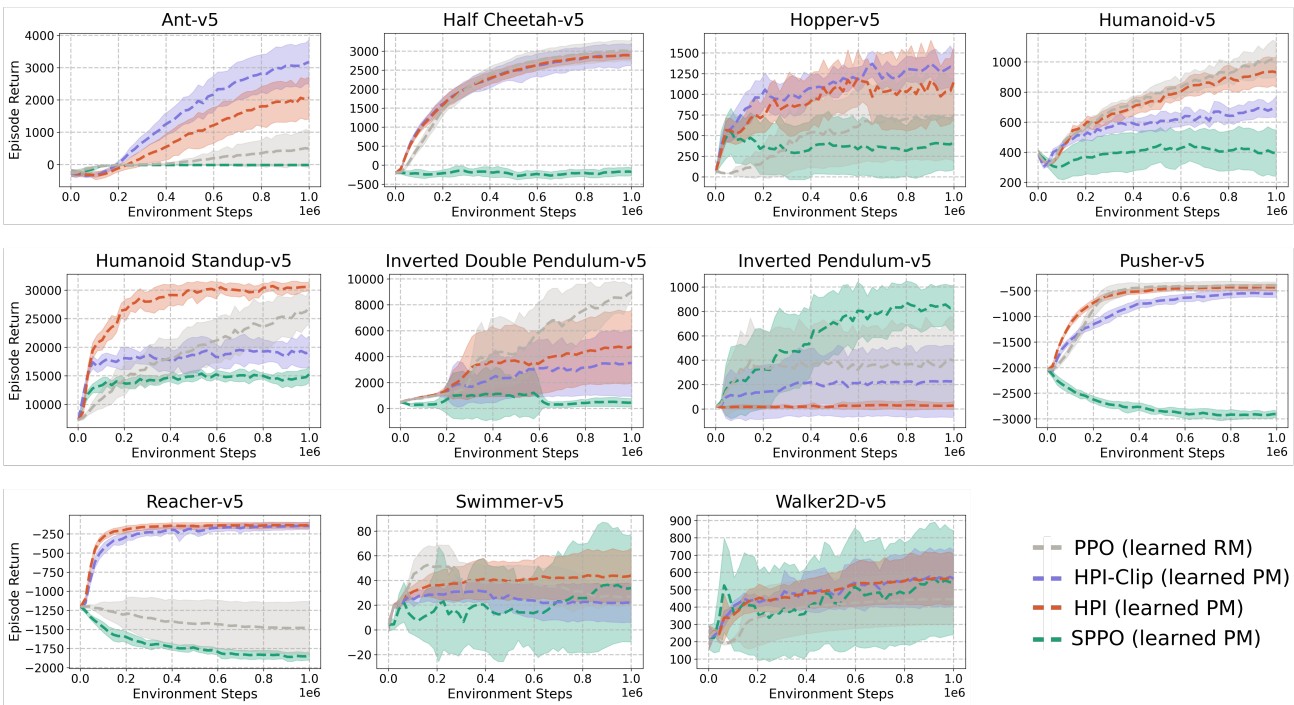

*Figure 5.* Comparison of algorithms that use learned preference models to attempt tasks on the Mujoco-v5 suite.

### A.3.2. MUJOCO-NT WITH LEARNED PREFERENCE MODELS

Figure 6 shows the performance of each algorithm that used a learned preference model to solve the Mujoco-v5 tasks.

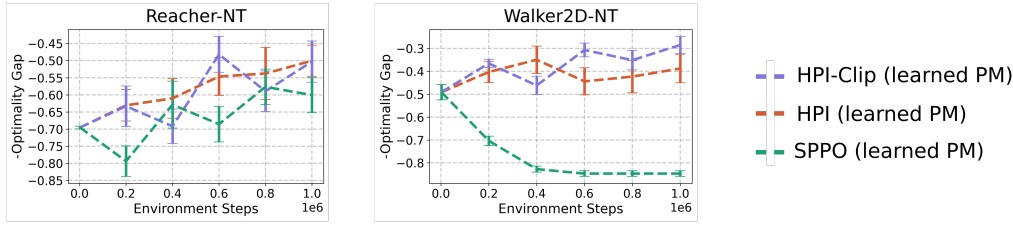

*Figure 6.* Comparison of algorithms that use learned preference models to attempt Reacher-NT and Walker2d-NT.

As above, we measured learning efficiency as the Area Under the Curve (AUC). The confidence interval for HPI-Clip's AUC was strictly higher than the confidence interval for SPPO's AUC in both Reacher-NT and Walker2d-NT Tasks. Similarly, the confidence interval for HPI's AUC was strictly higher than the confidence interval for SPPO's AUC on both tasks. Thus, we conclude that both HPI-Clip and HPI are more learning-efficient than SPPO on the Mujoco-NT.

### A.4. Ablation 2: Interpolating between Hedged Policy Iteration and Self-Play Preference Optimization

One important difference between HPI-Clip (Algorithm 4) and SPPO (Swamy et al., 2024, Algorithm 2) is that SPPO averages the per-timestep preference cumulant over the entire trajectory. This is because SPPO only receives a preference signal at the end of an episode. As a result of this averaging, SPPO may not evaluate policies correctly. HPI-Clip does not do this averaging. So we expect it to perform better.

We verify this by implementing variants of SPPO, where we vary the number of timesteps over which the per-timestep preference cumulant is averaged. We refer to this number of timesteps as the "comparison horizon" $T_C$. SPPO corresponds to the variant where $T_C$ is equal to the task horizon. In theory, HPI-Clip should correspond to the variant where $T_C = 1$. In Figure 7, we confirm that by varying the comparison horizon, we interpolate between HPI-Clip and SPPO. In most

environments, increasing the comparison horizon decreases performance.

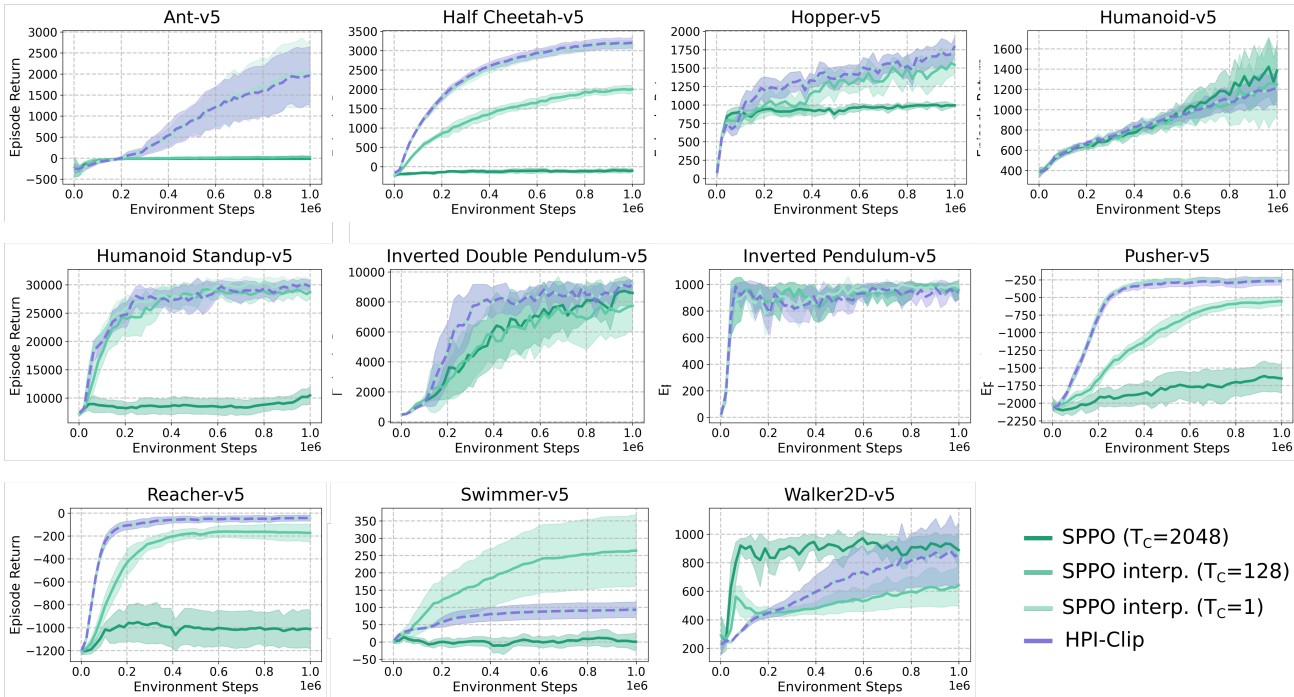

*Figure 7.* Interpolating between HPI-Clip and SPPO using the "comparison horizon" $T_C$, which is the number of timesteps over which per-timestep preference-cumulants are averaged.

## B. Related Work (Extended)

We elaborate on each of the column properties from Table 1 below.

**Reinforcement learning with pairwise preferences**   A problem model of reinforcement learning with pairwise preferences consists of a decision-making environment, a time horizon, and a preference margin. The domain of the preference margin has varied in prior work, and thus the optimality criteria have been different. Swamy et al. (2024); Wang et al. (2023); Chen et al. (2022), consider preference margins whose domain is $\mathcal{H}_N$, where $N \in \mathbb{N}$ is a finite time horizon. Gilbert et al. (2015; 2016) and Shani et al. (2024) consider preference margins whose domain is $\mathcal{S}$, and where preferences are given between final states (those which occur at timestep $N$). Wu et al. (2026) considers a preference margin defined between history-action pairs, which are given at each timestep $t$.

**Horizon.**    So far, problem models for reinforcement learning with pairwise preferences have considered settings where the problem ends at a known, finite time horizon. This may not be a good description of some important applications of reinforcement learning with pairwise preferences. People interacting with language models, for instance, typically do not end conversations after a fixed known timestep and reset to some initial distribution. Conversations may extend arbitrarily long, and beginnings and end steps may be unknown to the language model. Infinite-horizon models of sequential decision making handle these situations.

Infinite-horizon models of sequential decision making are also useful because they unify formalisms of tasks with finite and infinite horizons. As discussed by Sutton & Barto (2018, Chapter 3), finite horizon Markov decision processes can be represented as infinite-horizon problems. Thus, our infinite-horizon model, the Markov decision contest, can model applications of reinforcement learning with pairwise preferences with either finite or infinite horizons.

**Markov policies vs. history-dependent policies**   In Markov decision processes, restricting attention to Markov policies is justified because Markov policies perform optimally among the set of all history-dependent policies. This is useful for

applications, because it means that agents need only implement Markov policies in order to guarantee that they can perform just as well as any agent with a history-dependent policy.

Existing work on reinforcement learning with pairwise preferences have not considered guarantees on the performance of Markov policies relative to history-dependent ones. Chen et al. (2022); Wang et al. (2023); Swamy et al. (2024); Wu et al. (2026) consider history-dependent policies, but do not provide performance guarantees for Markov policies. On the other hand, Gilbert et al. (2015; 2016); Shani et al. (2024) consider only Markov policies, and do not discuss their performance relative to history-dependent ones. Here, we define our performance criteria with respect to history-dependent policies, but then show that Markov policies can be optimal. This extends the well-known result that Markov policies are optimal in infinite-horizon Markov decision processes under the average-reward criterion.

**Exact Solution Methods.** The problem of solving a finite unichain Markov decision process exactly is solvable in time polynomial in the number of states and actions (Puterman, 2014). Thus, it is said to be in P (Littman et al., 2013; Papadimitriou & Tsitsiklis, 1987).

Results on the complexity of solving reinforcement-learning problems that have pairwise preferences have not been provided yet. Here, we show that the problem of solving a Markov decision contest exactly is in P. This is not obvious. The generality of pairwise preferences with respect to reward function is known through logical axioms (Fishburn, 1982). It is not obvious from these axioms how much computational cost this generality brings.

**Approximate Solution Methods.** All work from Table 1 provide approximate solution methods with convergence guarantees. The convergence rates are all comparable to one another, as the problem models and optimality criteria in these models are differ.

Shani et al. (2024) and Wu et al. (2026) validate their approximate solution methods in experiments with function approximation. Swamy et al. (2024) experiment with function approximation using alternate version of their algorithm, but it requires approximations of history-dependent policies with Markov policies, which are not justified. The alternate version of Swamy et al. (2024)'s algorithm is discussed in our experiments. We did not compare with Shani et al. (2024) and Wu et al. (2026), because were developed for a regularized version of the problem.

## C. Proofs

### C.1. Proof of Lemma 4.3

**Lemma 4.3.** *When the transition probability function is unichain:*

1. $\Pi^{SR} \subset \Pi^1(\mu)$.

2. $X^1(\mu) = X^{SR}(\mu) = X$.

3. *$X$ is equal to the convex hull of $X^{SD}(\mu)$. In particular, $X$ is closed and convex.*

4. *For every state-action distribution $x \in X$, the stationary policy $\pi \in \Pi^{SR}$ given by the decision rule $d^\pi$, where*

$$d^\pi(a|s) = \begin{cases} \frac{x(s,a)}{\sum_{a'} x(s,a')} & \text{if } \sum_{a'} x(s,a') > 0 \\ \text{arbitrary} & \text{otherwise,} \end{cases}$$

*satisfies $x^{\pi,\mu} = x$.*

*Proof.* Part 1 follows from Proposition 8.9.1(a) of Puterman (2014). Parts (2) and (3) follow from slightly more general results that Puterman (2014) provides about subsequential limit points of average state-action frequencies. Puterman states these results in Theorem 8.9.4 and Corollary 8.9.5. We chose not to discuss subsequential limits here, to keep notation light. Part 4 follows from Corollary 8.8.7(a) of Puterman (2014). □

### C.2. Proof of Theorem 5.6

**Theorem 5.6.** *Within all finite unichain Markov decision contests:*

1. *There exists a randomized stationary policy that is optimal under the average-preference-margin criterion.*

2. *A randomized stationary policy is optimal under the average-preference-margin criterion if, and only if, it is a solution to*

$$\max_{\pi \in \Pi^{SR}} \min_{\pi' \in \Pi^{SR}} \sum_{s,a} \sum_{s',a'} x^\pi(s,a) x^{\pi'}(s',a') M((s,a),(s',a')). \tag{11}$$

To prove Theorem 5.6, we will use the following well-known results about two-player, zero-sum games played over the convex hull of a finite amount of points.

**Lemma C.1.** *For any closed, convex subset $C$ of $Dist(\mathcal{S} \times \mathcal{A})$,*

1. *The set of solutions to*

$$\max_{x \in C} \min_{x' \in C} \sum_{s,a} \sum_{s',a'} x(s,a) x'(s',a') M((s,a),(s',a')), \tag{12}$$

   *is non-empty.*

2. *An element $x^* \in C$ is a solution to (12) if and only if it satisfies*

$$\min_{x' \in C} \sum_{s,a} \sum_{s',a'} x^*(s,a) x'(s',a') M((s,a),(s',a')) \geq 0. \tag{13}$$

*Proof. (of Theorem 5.6)* First, we will show part (2) by showing that the average-preference-margin optimality condition can be expressed in the form (13). Then, we will use Lemma C.1 to prove part (1) of the theorem.

$$\pi^* \in \Pi^{SR} \text{ is optimal}$$

$$\overset{(a)}{\Longleftrightarrow} \inf_{\pi' \in \Pi^1} \sum_{s,a} \sum_{s',a'} x^{\pi^*}(s,a) x^{\pi'}(s',a') M((s,a),(s',a')) \geq 0$$

$$\overset{(b)}{\Longleftrightarrow} \inf_{x' \in X} \sum_{s,a} \sum_{s',a'} x^{\pi^*}(s,a) x'(s',a') M((s,a),(s',a')) \geq 0$$

$$\overset{(c)}{\Longleftrightarrow} \min_{x' \in X} \sum_{s,a} \sum_{s',a'} x^{\pi^*}(s,a) x'(s',a') M((s,a),(s',a')) \geq 0$$

$$\overset{(d)}{\Longleftrightarrow} x^{\pi^*} \text{ solves } \max_{x \in X} \min_{x' \in X} \sum_{s,a} \sum_{s',a'} x(s,a) x'(s',a') M((s,a),(s',a'))$$

$$\overset{(e)}{\Longleftrightarrow} \pi^* \text{ solves } \max_{\pi \in \Pi^{SR}} \min_{\pi' \in \Pi^{SR}} \sum_{s,a} \sum_{s',a'} x(s,a) x^{\pi'}(s',a') M((s,a),(s',a')).$$

(a) follows from the definition of the average-preference-margin criterion. (b) follows from Lemma 4.3, part (2). (c) follows from the fact that $X$ is closed and convex (Lemma 4.3, part (3)), and thus the function $x' \mapsto \sum_{s,a} \sum_{s',a'} x^{\pi^*}(s,a) x'(s',a') M((s,a),(s',a'))$, which is continuous, achieves its infimum in $X$. (d) is an immediate consequence of Lemma C.1 applied to $X$. (e) Follows from the fact that $X = \{x^\pi : \pi \in \Pi^{SR}\}$, by parts (1) and (2) of Lemma 4.3.

Part (1) of the theorem follows immediately from the first part of Lemma C.1 and relations (d) and (e).

$\square$

### C.3. Proof of Theorem 6.1

Recall the game from (6):

$$\max_{\pi \in \Pi^{SR}} \min_{\pi' \in \Pi^{SR}} \sum_{s,a} \sum_{s',a'} x^\pi(s,a) x^{\pi'}(s',a') M((s,a),(s',a')). \tag{14}$$

**Theorem 6.1.** *For every finite unichain Markov decision contest, there exists a linear program that solves the game in* (6) *using* $|\mathcal{S}||\mathcal{A}| + |\mathcal{S}| + 1$ *variables and* $2|\mathcal{S}||\mathcal{A}| + |\mathcal{S}| + 1$ *constraints.*

*Proof.* For clarity, we've numbered the steps in this proof.

(a) We can represent a policy $\pi \in \Pi^{SR}$ through its occupancy measure $x^{\pi}$. The decision rule

$$d^{\pi}(a|s) = \begin{cases} x^{\pi}(s,a) / \sum_{a'} x^{\pi}(s,a') & \text{if } \sum_{a'} x^{\pi}(s,a') > 0 \\ \text{arbitrary} & \text{otherwise} \end{cases}$$

determines $\pi$.

(b) By Lemma 4.3, $X = \{x^{\pi} : \pi \in \Pi^{SR}\}$. So, any solution to

$$\max_{x \in X} \min_{x' \in X} \sum_{s,a} \sum_{s',a'} x(s,a) x(s',a') M((s,a),(s',a')), \tag{15}$$

is an occupancy measure of an optimal stationary randomized policy. It remains to show that the game in (15) can be solved by a linear program with $|\mathcal{S}||\mathcal{A}| + |\mathcal{S}| + 1$ variables and $2|\mathcal{S}||\mathcal{A}| + |\mathcal{S}| + 1$ constraints.

(c) **Vector notation.** We will identify state-action distributions as vectors in $x \in \mathbb{R}^{|\mathcal{S}||\mathcal{A}|}$ whose entries are all nonnegative and sum to one (that is, $x \geq 0$ and $\mathbf{1}^T x = 1$). We will identify the preference margin $M$ as a matrix in $\mathbb{R}^{|\mathcal{S}||\mathcal{A}| \times |\mathcal{S}||\mathcal{A}|}$. So, for all state-action distributions $x$ and $x'$,

$$\sum_{s,a} \sum_{s',a'} x(s,a) x'(s',a') M((s,a),(s',a')) = x^T M x'.$$

So, in vector notation, the objective in (15) is

$$\max_{x \in X} \min_{x' \in X} x^T M x'.$$

(d) The $|\mathcal{S}|$ equations from (3) can be written as the system of linear equations

$$F_P x = 0,$$

where $F_P$ is the matrix $\mathbb{R}^{|\mathcal{S}| \times |\mathcal{S}||\mathcal{A}|}$ with $[F_P]_{s',s,a}$ equal to $P(s'|s,a) - 1$ if $s' = s$ and $P(s|s,a)$ otherwise. Now, $X$ can be written as the set of all vectors $x \in \mathbb{R}^{|\mathcal{S}||\mathcal{A}|}$ that satisfy the following conditions:

$$x \geq 0$$
$$\mathbf{1}^T x = 1$$
$$F_P x = 0.$$

(e) So, for every fixed $\tilde{x} \in X$, the following linear program (LP) solves the inner minimization problem, $\min_{x' \in X} \tilde{x}^T M x'$:

$$\begin{aligned} \text{minimize} \quad & \tilde{x}^T M x' \\ \text{subject to} \quad & x' \geq 0 \\ & \mathbf{1}^T x' = 1 \\ & F_P x' = 0. \end{aligned}$$

(f) The dual of this LP is

$$\begin{aligned} \text{maximize} \quad & \kappa \\ \text{subject to} \quad & M^T \tilde{x} + h^T F_P \geq \kappa \mathbf{1}. \end{aligned}$$

This dual program introduces two new variables: $\kappa \in \mathbb{R}$ and $h \in \mathbb{R}^{|\mathcal{S}|}$.

(g) By the duality theory of linear programs, the maximum value of the dual LP is equal to $\min_{x' \in X} \tilde{x}^T M x'$.

(h) So, the solution to $\max_{x \in X} \min_{x' \in X} x^T M x'$, maximizes the value of $\kappa$ for all possible $\tilde{x} \in X$. The following LP achieves this:

$$
\begin{aligned}
\text{maximize} \quad & \kappa \\
\text{subject to} \quad & M^T x + h^T F_P \geq \kappa \mathbf{1} \\
& x \geq 0 \\
& \mathbf{1}^T x = 1 \\
& F_P x = 0.
\end{aligned}
$$

This program has $|\mathcal{S}||\mathcal{A}| + |\mathcal{S}| + 1$ variables ($|\mathcal{S}||\mathcal{A}|$ from $x$, $|\mathcal{S}|$ from $h$, 1 from $\kappa$) and $2|\mathcal{S}||\mathcal{A}| + |\mathcal{S}| + 1$ constraints.

$\square$

The linear program for solving an average-reward Markov decision process with reward function $r : \mathcal{S} \times \mathcal{A} \to \mathbb{R}$ is:

$$
\begin{aligned}
\text{maximize} \quad & \sum_{s,a} x(s,a) r(s,a) \\
\text{subject to} \quad & F_P x = 0 \\
& \mathbf{1}^T x = 1 \\
& x \geq 0.
\end{aligned}
$$

In comparison to this linear program, the one we've used to solve Markov decision contests has an additional $|\mathcal{S}||\mathcal{A}|$ constraints (from its last inequality) and $|\mathcal{S}| + 1$ additional variables ($|\mathcal{S}|$ from $h$, and 1 from $\kappa$).

### C.4. Proof of Lemma 7.3

To prove Lemma 7.3, we need some additional definitions and results from Puterman (2014, Appendix A). For each stationary randomized policy $\pi \in \Pi^{SR}$, we define matrices $P_\pi$, $P_\pi^*$, and $H_{P_\pi}$ in $\mathbb{R}^{|\mathcal{S}| \times |\mathcal{S}|}$ through the following equations:

$$
[P_\pi]_{s',s} = \sum_a d^\pi(a|s) P(s'|s,a)
$$

$$
P_\pi^* = \lim_{T \to \infty} \frac{1}{T} \sum_{t=0}^{T-1} P_\pi^t
$$

$$
H_{P_\pi} = (I - P_\pi + P_\pi^*)^{-1}(I - P_\pi^*).
$$

Puterman calls $H_{P_\pi}$ the *deviation matrix*, in light of part 1 of the following lemma.

**Lemma C.2.** *When the transition probability function is unichain and aperiodic, then for all Markov decision rules $d^\pi$ and all states $s$ and $s'$*

1. $H_{P_\pi} = \sum_{t=0}^{\infty}(P_\pi^t - P_\pi^*)$. *In particular, the series $\sum_{t=0}^{\infty}([P_\pi^t]_{s',s} - [P_\pi^*]_{s',s})$ converges.*

2. $[P_d^*]_{s',s} = \nu^\pi(s')$, *where* $\nu^\pi(s) = \sum_a d^\pi(a|s) x^\pi(s,a)$.

3. *There exists a constant $\tau \geq 1$ such that $\sup_{\pi' \in \Pi^{SR}} \|H_{P_{\pi'}}\|_1 \leq \tau$.*

*Proof. (of Lemma C.2)* Part (1) is given as Theorem A.7(c) from Puterman's book. Puterman shows part (2) in Appendix A.4 of his book. It remains to show part (3). We've labeled the steps of the proof of part (3), for clarity.

(a) The set of all Markov decision rules can be represented as the $|\mathcal{S}|$-fold cartesian product of $\text{Dist}(\mathcal{A})$,

$$
(\text{Dist}(\mathcal{A}))^{|\mathcal{S}|} = \text{Dist}(\mathcal{A}) \times \cdots \times \text{Dist}(\mathcal{A}).
$$

Thus, the set of all Markov decision rules can be represented as a compact subset of $\mathbb{R}^{|\mathcal{S}||\mathcal{A}|}$. We denote this subset by $\mathcal{D}^{MR}$.

(b) The function $f : (\mathcal{D}^{MR}, \|\cdot\|_1) \to (\mathbb{R}, |\cdot|)$, given by $f(d) = \|H_{P_d}\|_1$, is continuous.

- This is because, when the transition probability function is unichain and aperiodic, the maps $d^\pi \mapsto P_\pi$ and $d^\pi \mapsto P_\pi^*$ are continuous, and the matrix $(I - P_\pi + P_\pi^*)$ is nonsingular. And so, $f$ continuous, as it is a composition of continuous functions.

(c) By (a) and (b), the function $f$ is a continuous function defined on over compact metric space. So, there exists a constant $\tau \geq 1$ such that

$$\sup_{\pi \in \Pi^{SR}} \|H_{P_\pi}\|_1 = \sup_{d \in \mathcal{D}^{MR}} f(d) \leq \tau.$$

$\square$

**Lemma 7.3** (Properties of marginal value functions). *If the Markov decision contest is aperiodic (in addition to being unichain), then there exists a constant $\tau \geq 1$ such that, for every stationary policy $\pi \in \Pi^{SR}$, state $s \in \mathcal{S}$, and action $a \in \mathcal{A}$,*

1. $|V_M^\pi(s)| \leq M_{max}\tau$ and $|Q_M^\pi(s, a)| \leq 2\, M_{max}\, \tau$.

2. $Q_M^\pi(s, a) = c_M^\pi(s, a) + \sum_{s'} P(s' \mid s, a) V_M^\pi(s')$.

3. $V_M^\pi(s) = \sum_{a'} d^\pi(a' \mid s) Q_M^\pi(s, a')$, where $d^\pi$ is the Markov decision rule that determines $\pi$.

*Proof.* Let $\tau$ be as defined in Lemma C.2. Once we establish that $V_M^\pi(s)$ is bounded, parts (2) and (3) follow immediately from the definitions of $V_M^\pi(s)$ and $Q_M^\pi(s, a)$. To show part (1), we will first show that $|V_M^\pi(s)| \leq M_{\max}\tau$, then use part (2) to show that $|Q_M^\pi(s, a)| \leq 2M_{\max}\tau$.

To show that $|V_M^\pi(s)| \leq M_{\max}\tau$, it suffices to show that

$$V_M^\pi(s) = \sum_{\tilde{s}, \tilde{a}} [H_{P_{d^\pi}}]_{\tilde{s}, s} \pi(\tilde{a}|\tilde{s}) c_M^\pi(\tilde{s}, \tilde{a}). \tag{16}$$

If (16) holds, then $|V_M^\pi(s)| \leq M_{\max}\tau$, since $\|H_{P_{d^\pi}}\|_1 \leq \tau$ and $|c_M(\tilde{s}, \tilde{a})| \leq M_{\max}$.

First note that, because $M$ is skew-symmetric,

$$0 = \sum_{\tilde{s}, \tilde{a}} \sum_{\tilde{s}', \tilde{a}'} x^\pi(\tilde{s}, \tilde{a}) x^\pi(\tilde{s}', \tilde{a}') M((\tilde{s}, \tilde{a}), (\tilde{s}', \tilde{a}')) = \sum_{\tilde{s}, \tilde{a}} x^\pi(\tilde{s}, \tilde{a}) c_M^\pi(\tilde{s}, \tilde{a}). \tag{17}$$

Therefore,

$$\sum_{\tilde{s}, \tilde{a}} [H_{P_\pi}]_{\tilde{s}, s} \pi(\tilde{a}|\tilde{s}) c_M^\pi(\tilde{s}, \tilde{a})$$

$$= \sum_{\tilde{s}, \tilde{a}} \left( \sum_{t=0}^{\infty} [P_\pi^t]_{\tilde{s}, s} - [P_\pi^*]_{\tilde{s}, s} \right) \pi(\tilde{a}|\tilde{s}) c_M^\pi(\tilde{s}, \tilde{a}) \qquad \text{(Apply Lemma C.2.(1))}$$

$$= \sum_{t=0}^{\infty} \sum_{\tilde{s}, \tilde{a}} ([P_\pi^t]_{\tilde{s}, s} - [P_\pi^*]_{\tilde{s}, s}) \pi(\tilde{a}|\tilde{s}) c_M^\pi(\tilde{s}, \tilde{a})$$

$$= \sum_{t=0}^{\infty} \sum_{\tilde{s}, \tilde{a}} ([P_\pi^t]_{\tilde{s}, s} - \nu^\pi(\tilde{s})) \pi(\tilde{a}|\tilde{s}) c_M^\pi(\tilde{s}, \tilde{a}) \qquad \text{(Apply Lemma C.2.(2))}$$

$$= \sum_{t=0}^{\infty} \sum_{\tilde{s}, \tilde{a}} ([P_\pi^t]_{\tilde{s}, s} \pi(\tilde{a}|\tilde{s}) - x^\pi(\tilde{s}, \tilde{a})) c_M^\pi(\tilde{s}, \tilde{a})$$

$$= \sum_{t=0}^{\infty} \sum_{\tilde{s}, \tilde{a}} [P_\pi^t]_{\tilde{s}, s} \pi(\tilde{a}|\tilde{s}) c_M^\pi(\tilde{s}, \tilde{a}) \qquad \text{(Apply (17))}$$

$$= \sum_{t=0}^{\infty} E_t^\pi [c_M^\pi(S_t, A_t)|S_0 = s]$$

$$= V_M^\pi(s).$$

This proves (16). The bound on $|Q_M^\pi(s,a)|$ follows by the triangle inequality, the bounds on $|c_M(s,a)|$ and $|V_M^\pi(s)|$, and the fact that $\tau \geq 1$:

$$\begin{aligned}
|Q_M^\pi(s,a)| &= \left| c_M^\pi(s,a) + \sum_{s'} P(s'|s,a) V_M^\pi(s') \right| \\
&\leq |c_M^\pi(s,a)| + \sum_{s'} P(s'|s,a) |V_M^\pi(s')| \\
&\leq M_{\max} + \sum_{s'} P(s'|s,a) M_{\max} \tau \\
&\leq 2 M_{\max} \tau.
\end{aligned}$$

$\square$

## C.5. Proof of Lemma 7.4

Recall that the average preference margin between policies $\pi, \pi' \in \Pi^1(\mu)$ is denoted by $\bar{M}(\pi, \pi')$:

$$\bar{M}(\pi, \pi') = \sum_{s,a} \sum_{s',a'} x^\pi(s,a) x^{\pi'}(s',a') M((s,a),(s',a')).$$

**Lemma 7.4** (Performance Difference Lemma). *If the Markov decision contest is aperiodic (in addition to being unichain), then, for all stationary policies $\pi, \pi' \in \Pi^{SR}$,*

$$\bar{M}(\pi, \pi') = \sum_{s,a} x^\pi(s,a)(Q_M^{\pi'}(s,a) - V_M^{\pi'}(s)).$$

*Proof.* By Lemma 4.3, $x^\pi \in X$, and so, by (3), it satisfies

$$\sum_{s,a} x^\pi(s,a) \sum_{s'} P(s'|s,a) V_M^{\pi'}(s') = \sum_{s,a} x^\pi(s,a) V_M^{\pi'}(s). \tag{18}$$

The expression for $\bar{M}(\pi, \pi')$ now follows from our previous definitions, Lemma 7.3.(2), and (18):

$$\begin{aligned}
\bar{M}(\pi, \pi') &= \sum_{s,a} \sum_{s',a'} x^\pi(s,a) x^{\pi'}(s',a') M((s,a),(s',a')) \\
&= \sum_{s,a} x^\pi(s,a) c_M^{\pi'}(s,a) \\
&= \sum_{s,a} x^\pi(s,a) \left( Q_M^{\pi'}(s,a) - \sum_{s'} P(s'|s,a) V_M^{\pi'}(s') \right) && \text{(Apply Lemma 7.3.(2))} \\
&= \sum_{s,a} x^\pi(s,a)(Q_M^{\pi'}(s,a) - V_M^{\pi'}(s')). && \text{(Apply (18))}
\end{aligned}$$

$\square$

## C.6. Proof of Theorem 7.5

**Theorem 7.5.** *Suppose that the Markov decision contest aperiodic (in addition to being unichain). When Hedged Policy Iteration (Algorithm 1) runs for $K \geq \log(|\mathcal{A}|)$ iterations and its learning rate $\eta$ is equal to $(2\,M_{max}\,\tau)^{-1}\sqrt{\log(|\mathcal{A}|)/K}$, the optimality gap its return policy is no greater than*

$$4\,M_{max}\,\tau \sqrt{\frac{\log(|\mathcal{A}|)}{K}}.$$

*Proof.* Hedged Policy Iteration (Algorithm 1) applies the Hedge algorithm (reviewed in Appendix G) to select the action probabilities of each state $s$ in parallel. For each state $s$, the Hedge algorithm score function at iteration $k$, denoted by $z_{s,k}$, is given by

$$z_{s,k}(a) = Q_M^{\pi_k}(s, a).$$

By Lemma 7.3, $|Q_M^{\pi_k}(s, a)| \leq 2M_{\max}\tau$, $\sum_a d^{\pi_k}(a|s)Q_M^{\pi_k}(s, a) = V_M^{\pi_k}(s)$, and the learning rate $\eta$ satisfies $|\eta\, z_{s,k}(a)| \leq 1$, for all $s$, $k$, and $a$. Thus, the Hedge algorithm bound (Theorem G.1) applied to each state $s'$ shows that,

$$\forall s' \in \mathcal{S}, \quad \max_p \sum_{k=1}^K \sum_a [p(a)Q_M^{\pi_k}(s', a) - V_M^{\pi_k}(s')] \leq \frac{\log(|\mathcal{A}|)}{\eta} + \eta\, K(2M_{\max}\tau)^2.$$

After dividing both sides of the inequality by $K$ and evaluating its right-hand side for our choice of $\eta$, the result shows that,

$$\forall s' \in \mathcal{S}, \quad \max_p \frac{1}{K} \sum_{k=1}^K \sum_a [p(a)Q_M^{\pi_k}(s', a) - V_M^{\pi_k}(s')] \leq 4M_{\max}\tau \sqrt{\frac{\log(|\mathcal{A}|)}{K}}.$$

So,

$$\max_{\pi \in \Pi^{SR}} \frac{1}{K} \sum_{k=1}^K \sum_{s,a} x^\pi(s, a)[Q_M^{\pi_k}(s, a) - V_M^{\pi_k}(s)] \leq 4M_{\max}\tau \sqrt{\frac{\log(|\mathcal{A}|)}{K}}.$$

But, the left-hand side of this inequality is equal to the exploitability of HPI's return policy, by the performance difference lemma (Lemma 7.4):

$$\max_{\pi \in \Pi^{SR}} \frac{1}{K} \sum_{k=1}^K \sum_{s,a} x^\pi(s, a)[Q_M^{\pi_k}(s, a) - V_M^{\pi_k}(s)] = \max_{\pi \in \Pi^{SR}} \frac{1}{K} \sum_{k=1}^K \sum_{s,a} \sum_{s',a'} x^\pi(s, a)x^{\pi_k}(s', a')M((s, a), (s', a')).$$

Therefore,

$$\max_{\pi \in \Pi^{SR}} \frac{1}{K} \sum_{k=1}^K \sum_{s,a} \sum_{s',a'} x^\pi(s, a)x^{\pi_k}(s', a')M((s, a), (s', a')) \leq 4M_{\max}\tau \sqrt{\frac{\log(|\mathcal{A}|)}{K}}.$$

$\square$

### C.7. Proof of Lemma 7.6

**Lemma 7.6.** *Let $x \in X$ be an occupancy measure. A stationary randomized policy $\pi \in \Pi^{SR}$ satisfies $x^\pi = x$ if and only if $\pi$ is given by a decision rule that is a solution to*

$$\max_\theta \sum_{s,a} x(s, a) \log d_\theta(a|s).$$

*Proof.* Suppose that $\pi$ is given by the decision rule $d^\pi$. The objective

$$\max_\theta \sum_{s,a} x(s, a) \log d_\theta(a|s)$$

is maximized by a decision rule $d^\pi$ if and only if, for all states $s'$ satisfying $\sum_a x(s', a) > 0$, $d^\pi(\cdot|s')$ is a solution to

$$\max_\theta \sum_a \frac{x(s', a)}{\sum_{a'} x(s', a')} \log d_\theta(a|s').$$

These objectives, in turn, are all simultaneously maximized by $d^\pi$ if and only if $d^\pi(a|s') = x(s', a)/\sum_{a'} x(s', a')$ for all actions $a$ and states $s'$ satisfying $\sum_{a'} x^{\pi'}(s, a') > 0$. This final condition is both necessary and sufficient to guarantee that $x^\pi = x$, by Lemma 4.3.(4). $\square$

## C.8. Proof of Proposition 7.7

The proof of Proposition 7.7 uses the following simple lemma.

**Lemma C.3.** *For every distribution $\nu \in Dist(\mathcal{S})$ and every function $f : \mathcal{S} \times \mathcal{A} \to \mathbb{R}$, if $d^* : \mathcal{S} \to Dist(\mathcal{A})$ is a solution to*

$$\max_{d:\mathcal{S}\to Dist(\mathcal{A})} \sum_{s,a} \nu(s)d(a|s)f(s,a)$$

*then, at all $s'$ for which $\nu(s') > 0$, $d^*(\cdot|s')$ is a solution to*

$$\max_{d(\cdot|s')} \sum_{a} d(a|s')f(s',a).$$

**Proposition 7.7.** *If the finite Markov decision contest is aperiodic (in addition to being unichain), then Algorithm 2 converges at the rate described in Theorem 7.5.*

*Proof.* To prove the proposition, we will need to separate the notation for HPI (Algorithm 1) and HPI PG (Algorithm 2). Let $d_k, \pi_k, x_k$ be the decision rule, policy, and occupancy measure of the $k$-th policy iterate of HPI. We will let $\hat{d}_k, \hat{\pi}_k, \hat{x}_k$ be the decision rule, policy, and occupancy measure of the decision rule at iteration $k$ of HPI PG. We will also let $\nu_k(s) = \sum_{a'} x_k(s, a')$. So $x_k(s, a) = \nu_k(s)d_k(a|s)$, and the HPI-PG decision rule $\hat{d}_{k+1}$ is chosen as a solution to

$$\max_{\theta} \sum_{s,a} \hat{\nu}_k(s,a)\hat{d}_\theta(a|s) \left( Q_M^{\hat{\pi}_k}(s,a) - \frac{1}{\eta} \log \frac{\hat{d}_\theta(a|s)}{\hat{d}_k(a|s)} \right). \tag{19}$$

Proposition 7.7 follows quickly if we can establish that:

(P) *For all iterations $k \geq 1$, actions $a' \in \mathcal{A}$, and states $s'$ that satisfy $\nu_k(s') > 0$,*

$$\hat{d}_k(a|s) = d_k(a|s).$$

If (P) holds, then, by Lemma 4.3.(4), $x_k = \hat{x}_k$ for all $k \geq 1$. Consequently,

$$\frac{1}{K} \sum_{k=1}^{K} \hat{x}_k = \frac{1}{K} \sum_{k=1}^{K} x_k.$$

Thus, by Lemma 7.6, the occupancy measure of the return policies of HPI and HPI-PG are equal. It then immediately follows that the optimality gap of the return policies of HPI and HPI-PG are also equal.

It remains to show (P). We do so by induction on the iteration $k$. We have labeled the steps, for clarity.

(a) The base case, when $k = 1$, is immediate because $\hat{d}_1(a|s) = d_1(a|s) = 1/|\mathcal{A}|$ for all $s$ and $a$.

(b) For the induction step, fix $k \geq 1$ and assume that, for all actions $a'$ and all states $s'$ satisfying $\nu_k(s') > 0$, $\hat{d}_k(a|s) = d_k(a|s)$.

(c) By Lemma 4.3.(4), $\hat{\nu}_k = \nu_k$. So, $\hat{d}_{k+1}$ is a solution to

$$\max_{\theta} \sum_{s,a} \nu_k(s,a)\hat{d}_\theta(a|s) \left( Q_M^{\hat{\pi}_k}(s,a) - \frac{1}{\eta} \log \frac{d_\theta(a|s)}{d_k(a|s)} \right).$$

(d) Because the environment is unichain, $Q_M^{\pi_k}(s,a) = Q_M^{\hat{\pi}_k}(s',a)$ for all actions $a$ and $s'$ satisfying $\sum_{a} x_k(s',a) > 0$. This is because the state-transition matrix of $\pi_k$ has a single recurrence class. So, when starting in a state $s'$ that satisfies $\nu_k(s') > 0$, every state $s''$ that is visited after $s'$ satisfies $\nu(s'') > 0$. Thus, the decision rules $\hat{d}_k$ and $d_k$ agree on every state visited after $s'$, and, as a result, $Q_M^{\pi_k}(s',a) = Q_M^{\hat{\pi}_k}(s',a)$.

(e) Thus, $\hat{d}_{k+1}$ is a solution to

$$\max_\theta \sum_{s,a} \nu_k(s,a) d_\theta(a|s) \left( Q_M^{\pi_k}(s,a) - \frac{1}{\eta} \log \frac{d_\theta(a|s)}{d_k(a|s)} \right).$$

(f) By Lemma C.3, for all $s'$ satisfying $\nu_k(s') > 0$, $\hat{d}_{k+1}(\cdot|s')$ is a solution to

$$\max_{d_\theta(\cdot|s')} \sum_a d_\theta(a|s') \left( Q_M^{\pi_k}(s',a) - \frac{1}{\eta} \log \frac{d_\theta(a|s')}{d_k(a|s')} \right).$$

(g) By the properties of the softmax update rule, $\hat{d}_{k+1}$ satisfies, for all actions $a'$ and all states $s'$ with $\nu_k(s') > 0$,

$$\hat{d}_{k+1}(a'|s') = d_k(a'|s') \exp(\eta\, Q_M^{\pi_k}(s',a')).$$

Thus, $\hat{d}_{k+1}(a'|s') = d_{k+1}(a'|s')$ for all actions $a'$ and states $s'$ satisfying $\nu_k(s') > 0$.

(h) Lastly, because the action probabilities of all policy iterates are strictly positive, $s'$ satisfies $\nu_k(s') > 0$ if and only if it satisfies $\nu_{k+1}(s') > 0$. So, $\hat{d}_{k+1}(a'|s') = d_{k+1}(a'|s')$ for all actions $a'$ and states $s'$ satisfying $\nu_{k+1}(s') > 0$.

(i) This completes the inductive step, and thus the proof of (P).

$\square$

## D. Relationship with the Bradley-Terry Model

The Bradley-Terry (BT) choice model, given by $\mathcal{P}_{\mathrm{BT}}((s,a),(s',a')) = \sigma(r(s,a) - r(s',a'))$, differs from the preference margin $M$ in the following sense: if $M((s,a),(s',a')) = \mathcal{P}_{\mathrm{BT}}((s,a),(s',a'))$ then the optimal policy under preference margin $M$ is not equal to the optimal policy under reward function $r$. Mathematically, the issue is that the set of solutions to $\max_\pi \min_{\pi'} \mathbb{E}_{\pi,\pi'}[\sigma(r(s,a) - r(s',a'))]$ is not equal to the set of solutions to $\max_\pi \mathbb{E}_\pi[r(s,a)]$.

Munos et al. (2024, Appendices A and B) argue that it is better to model preferences with a pairwise preference function instead of a BT model in many cases. But, of course, there may be cases where the input to the problem is truly stochastic choice data that is sampled from the BT model. Here, the difference is resolved by using the stochastic choice model $\mathcal{P}_M((s,a),(s',a')) = \sigma(M((s,a),(s',a')))$ to learn $M$ from the data. Under ideal conditions, the learned value for $M$ will be $r(s,a) - r(s',a')$. And then, for the reasons discussed in Section 5, an optimal policy of the Markov decision contest with preference margin $M$ will be an optimal policy of the Markov decision process with reward function $r$.

## E. Conditions on Transition Probability Functions

Here, we'll first discuss how finite-horizon Markov decision contests can be represented as infinite-horizon Markov decision contests. Then, we will elaborate on unichain and aperiodic transition probability functions.

### E.1. Representing Finite-Horizon Decision Problems as Infinite-Horizon Decision Problems

For $\gamma \in [0,1)$ and policy $\pi \in \Pi^{HR}$, the *$\gamma$-discounted occupancy measure of $\pi$*, denoted by $x^{\pi,\gamma}$, is given by

$$x^{\pi,\gamma}(s,a) = \sum_{t=0}^\infty \gamma^t \mathrm{Pr}_t^{\pi,\mu}(S_t = s, A_t = a).$$

Unlike average occupancy measures, the $\gamma$-discounted occupancy measures are well-defined for all transition probability functions and policies. Within finite Markov decision processes (MDPs), a policy is optimal under the discounted reward criterion if it maximizes the expected reward under its discounted occupancy measure (Sutton & Barto, 2018). In (Colaço Carr, 2026), we introduce the $\gamma$-discounted Markov decision contest, where optimality is defined similarly.

Sutton & Barto (2018, Chapter 3) show that any finite-horizon Markov decision process can be represented as a discounted, infinite-horizon Markov decision process. This is because the occupancy measure in the finite-horizon decision process can be associated with an occupancy measure in the infinite-horizon Markov decision process. For the same reasons, any finite-horizon Markov decision contest can be represented as a discounted, infinite-horizon Markov decision contest.

## E.2. Unichain and Aperiodic Transition Probability Functions

When the horizon is truly infinite, it is questionable whether state-action pairs should be prioritized depending on the order in which they occur. Sutton & Barto (2018, Chapter 10) argue that the parameter $\gamma$ should be removed from the reinforcement-learning problem definition, and that the average-reward criterion should be the default optimality criterion.

But, average reward is not always well-defined (Puterman, 2014). In average-reward MDP analysis, it is common to assume that the transition probability function is unichain. Under this assumption, the occupancy measures of all stationary policies are well-defined, and solutions can be recovered through linear programming.

Many reinforcement learning algorithms that solve average-reward Markov decision processes make use of *differential values* and *differential state-action values* $V_r^\pi(s)$ and $Q_r^\pi(s, a)$, which are given by the equations

$$V_r^\pi(s) = \sum_{t=0}^\infty E_t^\pi[r(S_t, A_t) - \bar{r}(\pi)|S_0 = s] \tag{20}$$

$$Q_r^\pi(s, a) = \sum_{t=0}^\infty E_t^\pi[r(S_t, A_t) - \bar{r}(\pi)|S_0 = s, A_0 = a]. \tag{21}$$

Here, $\bar{r}(\pi) = \sum_{s,a} x^\pi(s, a)r(s, a)$. Unfortunately, even when the transition probability function is unichain, the differential values may diverge. The aperiodicity assumption ensures that the differential values converge. And so, it is often assumed when analyzing reinforcement learning algorithms that solve average-reward MDPs (Sutton et al., 1999; Tsitsiklis & Van Roy, 1996). One simple way to ensure that a transition probability function is unichain and aperiodic is to add a small probability of returning to the initial state distribution in each state.

## F. Challenges with standard approximation methods

Recall the game from Equation 6:

$$\max_{\pi \in \Pi^{SR}} \min_{\pi' \in \Pi^{SR}} \sum_{s,a} \sum_{s',a'} x^\pi(s, a) x^{\pi'}(s', a') M((s, a), (s', a')).$$

As for the standard iterative solution for solving this game (online mirror descent), for $p \in \text{Dist}(\{1, \ldots, |\mathcal{A}|^{|\mathcal{S}|}\})$, define the policy $\pi_p \in \Pi^{SR}$ through its decision rule $d^{\pi_p}$, where

$$d^{\pi_p}(a|s) = \sum_{i=1}^{|\mathcal{A}|^{|\mathcal{S}|}} p(i) d^{\pi_{\text{det}}^i}(a|s).$$

The standard mirror descent method for solving this game would randomize $p_0$ as uniform, and then update according to

$$p_{k+1} \in \operatorname*{argmax}_p \bar{M}(\pi_p, \pi_{p_k}) - \frac{1}{\eta} \sum_{i=1}^{|\mathcal{A}|^{|\mathcal{S}|}} p(i) \log \frac{p(i)}{p_k(i)}. \tag{22}$$

This update step is challenging for two reasons. First, computing $\bar{M}(\pi_p, \pi_{p_k})$ is expensive. It requires computing the long-term average performance of policies $\pi$ and $\pi_k$ in an infinite-horizon decision process. Second, the regularization term (the second term) seems difficult to approximate. When the decision rule $d^{\pi_p}$ is parameterized by a weight vector, it's unclear how to estimate the probability $p(i)$ that the decision rule $d^{\pi_p}$ uses to "select" the deterministic policy $\pi_{\text{det}}^i$.

## G. The Hedge algorithm

The basic problem setting for the Hedge algorithm is as follows. At each iteration $k$, the algorithm selects a probability distribution $p_k$ over $n$ possible actions and observes a score $z_k : \{1, \ldots, n\} \to \mathbb{R}$. The Hedge algorithm initializes its first distribution $p_1$ as uniform random, and defines subsequent distributions $p_{k+1}$ according to the rule

$$p_{k+1}(i) \propto p_k(i) \exp(\eta z_k(i)).$$

Here, $\eta > 0$ is the learning rate. In Colaço Carr (2026), we review the Hedge algorithm and give a proof of the following result. The proof method was suggested by Arora et al. (2012). There are several variants of this convergence result, which use other conditions on the scores and learning rate (e.g. Cesa-Bianchi & Lugosi (2006); Kivinen & Warmuth (1997); Freund & Schapire (1995)).

**Theorem G.1** (Hedge bound). *If the Hedge Algorithm's learning rate $\eta$ is chosen so that $|\eta \, z_k(i)| \leq 1$ for all iterations $k$ and actions $i$, then its probability distributions $p_1, \ldots, p_k$ satisfy*

$$\max_p \sum_{k=1}^{K} \sum_{i=1}^{n} [p(i)z_k(i) - p_k(i)z_k(i)] \leq \frac{\log(n)}{\eta} + \eta \sum_{k=1}^{K} \|z_k^2\|_\infty.$$

*Here, $\|z_k^2\|_\infty = \max_i(z_k(i))^2$.*

*Proof.* See Colaço Carr (2026, Chapter 2). □

# H. Algorithm Details

Our code is available here: https://github.com/j-c-carr/markov-decision-contests

### H.1. HPI-Clip

For a randomized stationary policy $\pi \in \Pi^{SR}$, the *marginal advantage* $A_M^\pi(s,a)$ is defined as

$$A_M^\pi(s,a) = Q_M^\pi(s,a) - V_M^\pi(s).$$

Because the marginal advantage is equal to the marginal state-action value minus a state-dependent baseline, the decision rule updates of HPI (Algorithm 1) and HPI-PG (Algorithm 2) do not change if the marginal state-action values are replaced with marginal advantages. When this replacement is made for HPI-PG, the decision-rule update step becomes

$$\max_\theta \sum_{s,a} x^{\pi_k}(s,a) \frac{d_\theta(a|s)}{d_k(a|s)} \left( A_M^{\pi_k}(s,a) - \frac{1}{\eta} \log \frac{d_\theta(a|s)}{d_k(a|s)} \right). \tag{23}$$

This objective is maximized if $\frac{d_\theta(a|s)}{d_k(a|s)} A_M^\pi(s,a)$ is large and $\log \frac{d_\theta(a|s)}{d_k(a|s)}$ is small. Since $\log 1 = 0$, one surrogate objective for (23) is

$$\max_\theta \sum_{s,a} x^{\pi_k}(s,a) L_\epsilon^{\text{clip}}(s,a,d_\theta,d_k), \tag{24}$$

where

$$L_\epsilon^{\text{clip}}(s,a,d_\theta,d_k) = \min \left\{ \frac{d_\theta(a|s)}{d_k(a|s)} A_M^{\pi_k}(s,a), \text{clip}_{1\pm\epsilon} \left( \frac{d_\theta(a|s)}{d_k(a|s)} \right) A_M^{\pi_k}(s,a) \right\}. \tag{25}$$

Here, the function $\text{clip}_{1\pm\epsilon} : \mathbb{R} \to \mathbb{R}$ clips numbers to be within the range $[1-\epsilon, 1+\epsilon]$, and it effectively replaces the regularization term. When the decision-rule-update objective from HPI-PG is replaced with the objective in (24), we call the resulting algorithm *HPI-Clip*. This deep learning implementation of HPI-Clip is given in Algorithm 4.

When there exists a reward function $r$ such that $M((\tilde{s},\tilde{a}),(\tilde{s}',\tilde{a}')) = r(\tilde{s},\tilde{a}) - (\tilde{s}',\tilde{a}')$ for all $(\tilde{s},\tilde{a})$ and $(\tilde{s}',\tilde{a}')$, it is not too difficult to show that

$$A_M^\pi(s,a) = A_r^\pi(s,a),$$

where $A_r^\pi(s,a)$ is the advantage function used for solving Markov decision processes (Sutton & Barto, 2018). In this special case, the objective in (24) is equal to the PPO clip objective of Schulman et al. (2017, Equation 7). So, HPI-Clip can be viewed as a generalization of PPO, which learns from preference margins instead of reward functions.

---

**Algorithm 4** HPI-Clip (Deep Learning Implementation)

---

1: **Input**: clip parameter $\epsilon$, buffer $\mathcal{B}_{\text{avg}}$, number of behavior cloning steps $n_{\text{BC}}$,
2: Initialize decision rule $\hat{d}_1 : \mathcal{S} \to \text{Dist}(\mathcal{A})$ arbitrarily
3: Initialize marginal advantage estimate $\hat{A}_M^{\pi_1} : \mathcal{S} \times \mathcal{A} \to \mathbb{R}$ arbitrarily
4: **for** $k = 1, \ldots, K$ **do**
5:     # 1. Estimate occupancy measure
6:     Collect samples $\mathcal{B} = \{(s_t^i, a_t^i)_{t=0}^{T-1}\}_{i=1}^N$ from $\pi_k$
7:     $\mathcal{B}_{\text{avg}} \leftarrow \text{UpdateBuffer}(\mathcal{B}_{\text{avg}}, \mathcal{B})$
8:     # 2. Evaluate (approximately)
9:     $\hat{c}_{k,t}^i \leftarrow \frac{1}{|\mathcal{B}|} \sum_{s',a' \in \mathcal{B}} M((s_t^i, a_t^i), (s', a'))$
10:     $\hat{A}_M^{\pi_k} \leftarrow \text{UpdateAdv}(\hat{A}_M^{\pi_{k-1}}, \{(s_t^i, a_t^i, \hat{c}_{k,t}^i)_{t=0}^{T-1}\}_{i=1}^N)$
11:     # 3. Update decision rule
12:     Select $\hat{d}_{k+1}$ by optimizing:
13:     $\max_\theta \frac{1}{NT} \sum_{s_t^i, a_t^i} L_\epsilon^{\text{clip}}(s_t^i, a_t^i, d_\theta, \hat{d}_k)$
14: **end for**
15: # 4. Estimate average policy
16: Compute $\hat{d}_K^{\text{HPI}}$ by optimizing, for $n_{\text{BC}}$ gradient steps,
17: $\max_\theta \frac{1}{|\mathcal{B}_{\text{avg}}|} \sum_{s,a \in \mathcal{B}_{\text{avg}}} \log d_\theta(a|s)$
18: **Return** $\hat{d}_K^{\text{HPI}}$

---

## H.2. Implementation details

In light of the connections between HPI-Clip and PPO discussed in Section H.1, all algorithm implementations were based off of Huang et al. (2022)'s implementation of Proximal Policy Optimization. Both HPI-Clip and SPPO adapt PPO by supplying an estimate of the win-rate against the previous policy iterate as the reward function for the PPO update. We implemented SPPO Swamy et al. (2024), estimating the win-rate against the previous policy iterate as the sample average of the win-rate between the current state-action pairs and those stored in a queue of a fixed size $B$, which gets updated at each iteration. The critical difference for SPPO is that, because it is designed to optimize for preferences between trajectories, it takes the per-timestep reward as the trajectory average. In Appendix A.4 we study the relationship between HPI-Clip and SPPO in greater depth.

HPI was implemented in the exact same way as HPI-Clip, except that PPO's clipped surrogate objective was instead replaced with the objective from Algorithm 2. For HPI, we used a learning rate of $\eta = 1.5$ across all tasks.

**Behavior cloning.** To estimate the occupancy-measure-matching policies of HPI and HPI-Clip, we performed either 0 or 20 epochs of stochastic gradient ascent to the objective from Lemma 7.6—this we call the behavior cloning step. We began behavior cloning from the final policy iterate of these algorithms and used a stored buffer $\mathcal{B}_{\text{avg}}$ of sample trajectories to estimate the distribution $\frac{1}{K} \sum_{k=1}^K d^{\pi_k}$. In our experiments, we stored one trajectory per iteration (which amounted to roughly 490 trajectories stored in total), though in future work it would be interesting to explore how to reduce the number of samples required in this step. We did zero steps of behavior cloning on the Mujoco suite and 20 steps of behavior cloning in tasks with nontransitive preferences.

## H.3. Hyperparameters and network architectures

**Shared hyperparameters.** Table 2 shows the list of hyperparameters shared across HPI-Clip, PPO, and SPPO. HPI replaced the "Clip epsilon" hyperparameter with the learning rate $\eta$. We explored different hyperparameter choices for three hyperparameters: the learning rate $\eta$ of HPI, the queue size $B$, and the learning rate for Adam in SPPO. We did a hyperparameter search for HPI's learning rate $\eta$ in $\{0.75, 1.5, 3, 6, 12\}$ by observing performance in Ant-v5, HalfCheetah-v5, and Inverted Double Pendulum-v5 and taking the best average performance (average reward over the last 100,000 environment steps) over five random seeds.

The HPI-Clip, SPPO, and HPI algorithms all had an additional hyperparameter for the queue size. In the Mujoco-v5 suite, we started with a queue size of $B = 100$, as was suggested in Swamy et al. (2024). However, we found that all algorithms

*Table 2.* Shared hyperparameters. From (Huang et al., 2022).

| Parameter | Value |
|---|---|
| Learning rate | $3 \times 10^{-4}$ |
| Adam epsilon (numerical stability for optimizer) | $1 \times 10^{-5}$ |
| Environment steps per policy update | 2048 |
| Total environment steps | $1 \times 10^{6}$ |
| Update Epochs | 10 |
| Number of minibatches | 32 |
| GAE gamma | 0.99 |
| GAE lambda | 0.95 |
| Clip epsilon | 0.2 |
| Entropy coefficient | 0.0 |
| Value function coefficient | 0.5 |
| Max gradient norm | 0.5 |
| Activation function | `tanh` |
| Anneal learning rate | `True` |

would suffer from performance collapse in easy tasks, such as Inverted Pendulum-v5, where it quickly becomes almost impossible for the policy iterates to "win" over previous iterates when previous iterates are consistently near-optimal. There are several methods for mitigating this (e.g. by applying early stopping); we simply chose to store and compare against an additional 100 random samples from the first policy iterate across all timesteps, and that resolved the issue. As in Swamy et al. (2024), we reduced the queue size to $B = 10$ for tasks with non-transitive preferences.

The final hyperparameter we considered was learning rate for SPPO's Adam optimizer. Swamy et al. (2024) increase the learning rate with respect to their baseline algorithm (Soft Actor Critic) tenfold. However, we found that increasing the learning rate of default PPO decreased performance of in all three environments in which we conducted hyperparameter tuning (Ant-v5, HalfCheetah-v5, and Inverted Double Pendulum-v5). The discrepancy here is likely due to the fact that the reinforcement learning algorithm implemented to update the policy in Swamy et al. (2024)'s paper was Soft Actor Critic (Haarnoja et al., 2018), while here we chose to implement the reinforcement learning algorithm update with Proximal Policy Optimization (Schulman et al., 2017), because of the relationship discussed in Appendix H.1. After noting this discrepancy for SPPO, we did not try changing the learning rate for HPI-Clip from Huang et al. (2022)'s default value for PPO.

**Behavior cloning hyperparameters.** The BC training uses a higher learning rate ($1 \times 10^{-2}$ vs $3 \times 10^{-4}$ for PPO) and relaxed gradient clipping (1.0 vs 0.5). The objective minimizes the negative log-probability of the actions in $\mathcal{C}$ under the current policy distribution. The trajectories in $\mathcal{C}$ are flattened, shuffled, and processed in minibatches over multiple epochs. Table 3 shows the full list of behavior-cloning hyperparameters.

To choose the hyperparameters for behavior cloning, we searched for the BC learning rate in $\{1 \times 10^{-2}, 1 \times 10^{-3}\}$ and the number of update epochs in $\{0, 20, 50\}$ by choosing those that led to the highest performance after 400 000 training steps in Reacher-NT (we used a shorted amount of training steps, because evaluation on non-transitive preferences is significantly more computationally expensive).

*Table 3.* Behavior cloning (BC) hyperparameters, used for HPI and HPI-Clip only.

| Parameter | Value | Description |
|---|---|---|
| BC Number of epochs | 20 | Number of full passes through BC data |
| BC Learning rate | $1 \times 10^{-2}$ | Learning rate for BC optimizer |
| BC Number of minibatches | 32 | Number of minibatches per epoch |
| BC Max gradient norm | 1.0 | Gradient clipping threshold |

**Network architectures.** The network architectures used to represent the policy and value networks were kept constant across all algorithms and were the default ones used in Huang et al. (2022)'s implementation of PPO. The policy and value networks share an identical two-layer fully-connected architecture with 64 hidden units each, using Tanh activation functions and orthogonal weight initialization with scaling factors of $\sqrt{2}$ for hidden layers. The policy network outputs continuous actions through a multivariate normal distribution with diagonal covariance, where the mean is produced by a final dense layer (orthogonally initialized with 0.01 scaling) and the log standard deviation is a learnable parameter shared across environments but separate for each action dimension. The value network uses the same hidden layers but terminates with a single scalar output (orthogonally initialized with unit scaling) to estimate state values.

### H.4. Pseudocode for Mujoco-NT Preference Margins

Here is how the preference margins for ReacherNT and Walker2dNT were implemented:

```
def reacher_nt_preference(obs_1, obs_2):
    radius_pref = 2 * ((obs_1.radius > obs_2.radius) - 0.5)

    difference = math.fmod(obs_1.angle + angle/2.0 - obs_2.angle, 2 * math.pi)
    angle_pref = difference < theta/2.0 or difference > 2 * math.pi - theta/2.0
    angle_pref = 2 * (angle_pref - 0.5)
    return 0.3 * radius_pref + 0.7 * angle_pref

def walker2d_nt_preference(obs_1, obs_2):
    height_1 = clip((obs_1[0] - 1.0) / 0.3, 0, 1)
    height_2 = clip((obs_2[0] - 1.0) / 0.3, 0, 1)

    speed_1 = clip(obs_1[8] / 2.0, 0, 1)
    speed_2 = clip(obs_2[8] / 2.0, 0, 1)

    stability_1 = clip((-|obs_1[1]| + 0.5) / 0.5, 0, 1)
    stability_2 = clip((-|obs_2[1]| + 0.5) / 0.5, 0, 1)

    # Find dominant feature
    # 0=height, 1=speed, 2=stability
    dominant_1 = argmax([height_1, speed_1, stability_1])
    dominant_2 = argmax([height_2, speed_2, stability_2])

    # Preference matrix:
    #           High  Fast  Stable
    # High   [   0,    1,    -1  ]
    # Fast   [  -1,    0,     1  ]
    # Stable [   1,   -1,     0  ]

    return preference_matrix[dominant_1, dominant_2]
```

