# OpenReview forum: "Reinforcement Learning with Pairwise Preferences in Long-Term Decision Problems"
_ICML.cc/2026/Conference — ICML 2026 regular_

### Official Review · Reviewer_z3tX · 2026-03-06

**Soundness:** 3
**Presentation:** 2
**Significance:** 4
**Originality:** 3
**Overall Recommendation:** 6
**Confidence:** 2

**Summary:**

**summary:** The paper considers the problem of learning an optimal policy (which beats or ties with all other policies) for agents whose actions are modeled by a Markov decision process  with an infinite decision horizon. They consider a scenario when rather than having access to a reward function, the agents use pairwise preferences for finding the optimal policy.
They introduce  *"Markov decision contest"* which generalizes finite decision models to infinite horizons, and they show that in this model optimal policy exists, and can be found using an LP  with a finite number of variables. Finally, they introduce HPI, an iterative algorithm that converges to an optimal policy at a sub-linear rate.

**significance** Finding an optimal policy in an infinite MDP is a tractable problem when rewards are known but (as the paper claims) has been intractable when pairwise preferences are used. Preference-based learning is increasingly relevant in reinforcement learning, particularly in applications motivated by large language models. For this reason, the results presented in the paper could have significant impact on the community (although my primary expertise is outside RL).

Overall, the paper is mathematically sound, and the contribution represents a solid theoretical advance supported by realistic experiments.


**presentation**

- clarity: I find the presentation of the paper a bit confusing, especially for someone outside the RL community. *"Learning from pairwise preferences"* refers to a different problem in theoretical CS and OR community:

1- TCS: A true ordering of objects is known but users have access to noisy pairwise samples and the goal is to retrieve the true order see e.g.,
"Noisy sorting without resampling" M. Braverman, Elchanan Mossel, SODA 2018,
"Simple, Robust and Optimal Ranking from Pairwise Comparisons" Nihar B. Shah, Martin J. Wainwright, JMLR 2018

2- OR: pairwise preferences are drawn from a probabilistic model for instance Mallows model and the goal is to learning the parameters of this model, See e.g.,
"Mallows ranking models: maximum likelihood estimate and regeneration" Wenpin Tang ICML 2019,
"Probabilistic Preference Learning with the Mallows Rank Model" Vitelli et al. JMLR 2018.,
"Generalized Top-k Mallows Model for Ranked Choices" Shahrzad Haddadan, Sara Ahmadian ICML 2025,
"On A Mallows-type Model For (Ranked) Choices" Yifan Feng, Yuxuan Tang, NeurIPS 2022.


I understand that this is a reinforcement learning paper, but the introduction does not mention reinforcement learning or Markov Decision Processes until the last paragraph. This was quite confusing for me, and the context only became clear around page 3 when the related work was discussed.

I encourage the authors to revise the introduction to clarify the context earlier. For example, it could start with a sentence such as *"In the reinforcement learning literature…"* or something similar that immediately clarifies that the problem is within the RL framework. It would also be helpful to include a brief explanation (one or two sentences) contrasting the proposed problem with the classical formulations in the field, especially for readers who may not be part of the RL community.

- notation: I find the notation a bit heavy but I don't see any way around it since it is a highly theoretical paper.  The words *"agent"* and *"policy"* are sometimes used ambiguously.  For instance in the first sentence of the introduction it helps if you say you are looking for an optimal policy.

- typos and general structure:  I would put the summary of contributions after related work to highlight and contrast the distinctions. There are a few typos but not too many:

First column: ever-> every
Page 3 first column: mixing time definition: extra "the"

**Compliance With Llm Reviewing Policy:**

Affirmed.

**Final Justification:**

**Disclaimer:** I am not a member of the LR community, though I have some familiarity with the literature in this area.

As mentioned earlier, I find the paper theoretically sound and addressing an important problem. In my initial assessment, I found parts of the presentation a bit unclear. The authors have indicated that they will use the additional page in the camera-ready version to improve clarity and address these issues.

While I would not place this paper among the top 10% of those I have read, I consider it to be high quality and definitely above the acceptance threshold for ICML.

**Key Questions For Authors:**

In definition 4.2 you are defining a **finite** Markov decision contest. How is this helping in the analysis of **infinite** trajectories?

**Limitations:**

yes

**Strengths And Weaknesses:**

The strengths of the paper are
 1- the importance of the problem. The analysis of infinite or exponentially sized MDP s are useful for applications in LLM when the trajectory of the MDP is often extremely large.

2- Solid mathematical analysis of the problem

I think the main weakness is the presentation specially for readers outside RL community

---

> ### Author Rebuttal · Authors · 2026-03-30
>
> Thank you for your comments. We’re pleased to hear that you think the paper’s contributions represent a solid theoretical advance that could have a significant impact on the community.
>
> A summary of contributions after the related work is a good idea; we will add one. Our plan to integrate your other suggestions into a revised paper is discussed below.
>
> **[Context for RL/pairwise preferences]**
>
> Clarifying context early in the introduction is a great suggestion. The introduction for the submission was cut short due to space constraints. Here’s how we'll update it, in light of your suggestions:
> 1. In the introduction and abstract, we’ll change the word “agent” to “reinforcement learning agent”. So the first words of the abstract will be “Reinforcement learning agents that can beat or tie…”, to signal the context early.
> 2. We will add a sentence in the first paragraph to explain that, in our paper, “learning from pairwise preferences” means finding a policy that corresponds to the Nash equilibria of a skew-symmetric preference function.
> 3. Below the first paragraph of the introduction, we will add a paragraph to contrast this approach to the classical formulation of reinforcement learning from human feedback from Casper (2025) and Christiano (2017). We’ll also discuss these classical formulations in the related work section.
>
> **[Agent vs. policy]**
>
> We will edit the paper to disambiguate between agents and policies. In the preliminaries, we’ll clarify that an agent is the decision-maker, whose goal is to find an optimal policy. And we’ll include a reference to Sutton and Barto (2018, Chapter 3.1)’s section on agents.
>
> **[Finite vs. infinite]**
>
> The Markov decision contest is called finite because there are a finite number of states and actions. The analysis is on infinite trajectories, which are infinite sequences of alternating states and actions. A large language model with a finite vocabulary of tokens, but which runs for an infinite number of timesteps, would be an example of such a contest.
> This convention is also adopted in Sutton and Barto (2018, see Chapter 3). They define a finite Markov decision process as one with a finite number of states and actions, and within it they analyze infinite trajectories. We will clarify this convention in the text, below the definition of a Markov decision contest on page 4.
>
> Casper, Stephen, et al. "Open Problems and Fundamental Limitations of Reinforcement Learning from Human Feedback." Transactions on Machine Learning Research (2025).
>
> Christiano, Paul F., et al. "Deep reinforcement learning from human preferences." Advances in neural information processing systems 30 (2017).
>
> Sutton, R. S., & Barto, A. G. (2018). Reinforcement Learning: An Introduction (Second ed.). The MIT Press

---

> > ### Author Rebuttal · Reviewer_z3tX · 2026-04-03
> >
> > The authors have adequately addressed my questions, confirming my initial assessment that this is a strong paper. Assuming the revisions are made as indicated, I am increasing my score.

---

> > > ### Author Response · Authors · 2026-04-06
> > >
> > > Thank you for your consideration. We’re excited to update the paper with your suggestions and those of the other reviewers.

---

### Official Review · Reviewer_SHcU · 2026-03-08

**Soundness:** 3
**Presentation:** 1
**Significance:** 2
**Originality:** 2
**Overall Recommendation:** 2
**Confidence:** 1

**Summary:**

In this paper, authors propose to learn from pairwise preferences in the context of long term decision making problems.  The authors propose "Markov Decision Contest" which in my understanding is presented as a generalisation of MDPs. Authors claim that agents only need a stationary Markov policy in order to be optimal and its possible to achieve such a policy in polynomial time $(|A|^{|S|})$.  The authors also propose algorithms that achieve such a policy with sublinear convergence.

**Compliance With Llm Reviewing Policy:**

Affirmed.

**Key Questions For Authors:**

I think the biggest assumption you make is that state and actions (of agents) are sufficient to describe pairwise preferences, is it always true ? Aren't you assuming that the past does not matter? Is it to simplify the math or are there are solid reasons behind it. Perhaps you may want to elaborate on this ? The travel agent booking example (4.3) that is written in the paper actually exemplifies the simplicity of the assumption that may break in the real world.  Perhaps I am missing something, but I believe this is a key point in the paper and an elaboration here will help.

**Limitations:**

Authors have not outlined the limitations of their work.  I dont think the authors discuss many applications to get into details of potentially negative impacts. The work is more theoretical in nature.

**Strengths And Weaknesses:**

I have tried to read the paper many times and specially the key theorems -- 4.7 and 4.8, before I could fight the introduction section. The manuscript is very difficult to follow.  Even the first three sentences of the paper are laden with typos (or a missed words here and there). Here is a sample: "When learning from pairwise preferences, a particularly strong notion of success is achieved by an agent can beat
or tie ever other agent in a pairwise matchup".  Similarly just the third sentence of the paper is literally a paragraph.
The problem of learning from pairwise preferences is an interesting one and the theorems 4.7 and 4.8 are promising, but the articulation of the approach can be much better. The writing of the paper can be much improved and that is the strongest criticism of this work.

The strength of the work revolves around theorems 4.7 and 4.8.  The framing of Markov Decision Contest (MDC) like a generalisation of MDP(?) is probably novel via a pariwise comparison of agents is the strength and the key idea of the paper. If one takes away the requirement of "infinite" horizon for optimality, it opens up interesting applications (which the authors have not elucidated on).

---

> ### Author Rebuttal · Authors · 2026-03-30
>
> Thank you, we appreciate your feedback. We’re glad that you found the problem interesting and the theoretical results promising.
>
> We want the revised manuscript to be as easy to read as possible. We will correct typos, and we will cut sentences that are more than three lines long. In our response to Reviewer z3tX, we’ve outlined three specific changes that we’ll make to the introduction,  for improved clarity. Our responses to Reviewers xuxx and Xqiw contain several other details and explanations that will be included in our revised paper, which may address some of your concerns.
>
> Below, we’d like to make one clarification about the summary you provided, and then we’ll respond to your other comments.
>
> **[Clarification on summary]**
>
> > “Authors claim [...] its possible to achieve [an optimal policy] in polynomial time ($|A|^{|S|}$)."
>
> We show that it’s possible to achieve an optimal policy in time *polynomial in $|S||A|$, not $|A|^{|S|}$*. The difference is significant because it changes the complexity class of the problem. With an algorithm that runs in time polynomial in $|S||A|$, we've shown that the problem of solving a Markov decision contest is in P. The runtime is discussed in the paragraph before Theorem 4.8, and it is stated in the theorem.
>
>
> **[Sufficiency of states and actions to describe preferences]**
>
> Elaborating on our assumption that preferences can be described in terms of states and actions is a great suggestion. We actually don’t think that this assumption is strong, as it is the same one used in Markov decision processes, where the reward function is defined in terms of states and actions.
>
> The sufficiency of state-action pairs to describe preferences may seem like a strong assumption for a fixed set of states (say, if the set of states is the set prompts you could give to the travel-booking agent). But one can always expand the set of states to make it more descriptive (say, by expanding the state set to include the last $N$ prompts, for some large number $N$). Of course, this will make the set of states and actions very large. But, for MDPs, deep reinforcement learning methods can effectively solve problems with very large state and action spaces. We expect that it will be possible to develop similarly scalable approaches to solve Markov decision contests.
>
> **[Applications of infinite-horizon models]**
>
> > “If one takes away the requirement of "infinite" horizon for optimality, it opens up interesting applications.”
>
> We agree that applications merit more discussion (which we'll add). But we think that the infinite-horizon model *opens up new applications*, rather than taking away from them. The reason is that our model can describe all tasks with finite time horizons, while also modeling tasks with variable and unknown time horizons, which are not described with existing finite-horizon methods. Please see our response to Reviewer Xqiw for more details. We’ll add this discussion to our paper.
>
>
> We hope that our response, along with the improvements we’ve committed to in our responses to other reviewers, address your main concerns. If you have other suggestions for improving the paper, please let us know.

---

> > ### Author Rebuttal · Reviewer_SHcU · 2026-04-03
> >
> > I am assuming that the authors will increase the readabiity of the paper. I am increasing the score by 1 grade.

---

> > > ### Author Response · Authors · 2026-04-06
> > >
> > > Thank you—increasing readability will be a priority during our revisions.

---

### Official Review · Reviewer_xuxx · 2026-03-12

**Soundness:** 3
**Presentation:** 4
**Significance:** 3
**Originality:** 3
**Overall Recommendation:** 5
**Confidence:** 3

**Summary:**

# summary
This paper studies sequential decision problems when only pariwise preferences are available. This paper formulates the problem as an Markov decision contest, i.e., an MDP with a Markov preference model that depends on state and action pairs. It then establishes the following two key claims:
    - The problem of solving for optimal policies for Markov decision contest is in P.
    - There exists an optimal stationary Markov policy that beats or ties all history dependent policies.
Building upon the above facts and necessary assumptions, the paper then present policy iteration algorithms and provide their theoretical analysis when the state and action spaces are finite, showing a convergence rate proportional to $\sqrt{1/K}$ where $K$ is the number of iterations.

The algorithm is then extended to large state and action spaces with function approximation, with the performance evaluated in the simulations.

**Compliance With Llm Reviewing Policy:**

Affirmed.

**Final Justification:**

I am raising the score because all my concerns are well addressed in the rebuttal.

**Key Questions For Authors:**

Please address my concerns and questions in the weakness section. I am willing to raise the score if the major weaknesses are properly addressed.

**Limitations:**

Yes.

**Strengths And Weaknesses:**

# Strengths
1. The paper is very well written with minor typos.
2. The paper develops deep and sound theoretical analysis of the proposed problem, Markov decision contests, including solvability, optimality of stationary Markov policies, and convergence of the policy iteration algorithms.
3. The proposed algorithms are evaluated in various simulation environments.

# Major Weakness
1. Evaluating the $Q$ function. $Q$ evaluation requires knowing either the function $\phi$ or the $M_{\phi}(\delta^{(s,a)},d^{\pi})$. I encourage the authors to at least extend the experiment results to the setting where the preference function $\phi$ is unknown and learned from data.
2. Calculating output policy. The currently algorithm outputs an occupancy-measure-matching policy. Calculating such policies requries all policies during iteration $\{d^{\pi_k}\}_{k\in[K]}$, or at least an average of the policies $\sum_{k=1}^K d^{\pi_k}$. Is there a memory efficient way to achieve this without saving all policies, especially when the state and action spaces are large?
3. The authors haven't included discussions on the relation between the studied preference model $\phi(s,a)$ with other existing models, e.g., the Bradley-Terry preference model for binary choices. It is recommended to discuss whether the studied preference model covers those exisitng models or they are fundamentally different.


# Other Weakness
1. Missing details. The authors defer details of policy evaluation (for Algorithm 1, 2, and 3) as well as the algorithm with function approximation to the appendices. They are important for understanding the algorithm and implmentation and deserves space in the main text.
2. Unclear output policy. Do algorithm 2 and 3 also output occupancy-measure-matching policies, or they directly output the last-iterate policy?
3. Typos. For example, Equation (13) at the start of Section 5 should be Equaiton (5).
4. It might be better to present a high-level on "standard interative algorithms to solve Markov decision contest" (Section 5).

---

> ### Author Rebuttal · Authors · 2026-03-31
>
> Thanks for your thoughtful feedback. We’ll first address the major weaknesses and then your other points. We'll integrate all of these comments into a revised paper.
>
> **[M1: Experiments with learned preference model]**
>
> These experiments were almost done before our submission and are now finished.
>
> To learn a reward model, we followed the protocol of Christiano (2017). The learned preference model used the same protocol, except that the preference model network took two state-action pairs as input, instead of one.
> The reward model was used to train a PPO agent, and the preference model was used to train HPI, HPI-Clip, and SPO agents. The performance of these agents reward-based tasks, in terms of mean return over last 100k environment steps is given below, with standard error bars over five random seeds:
>
> | | PPO+RLHF | HPI | HPI-Clip | SPO |
> | --- | --- | --- | --- | --- |
> | Ant-v5 | 1201$\pm$ 502 | $0\pm 0$ | $1876\pm 511$ | $0\pm 0$ |
> | InvertedDoublePendulum-v5 | $9040\pm 439$ | $8134\pm 797$ | $8912\pm 376 $ | $8095  \pm 650 $ |
> | Reacher-v5 | $-189\pm 66$ | $-145\pm 81$ | $-150\pm 63$ | $-1000\pm 224$ |
> | Pusher-v5 | $-497\pm 28$ | $-450\pm 40$ | $-519\pm 35$ | $-1676\pm 310$ |
>
> The performance in tasks with nontransitive preference, in terms of exploitability after 1M env. steps (with standard error bars across 5 random seeds) was:
>
> | | HPI | HPI-Clip | SPO (best) | SPO (last) |
> | --- | --- | --- | --- | --- |
> | Reacher-NT | $-0.48\pm 0.05$ | $-0.48\pm 0.06$ | $-0.54\pm0.05$ | $-0.62\pm 0.06$ |
> | Walker2d-NT | $-0.40 \pm 0.07$ | $-0.30\pm 0.05$ | $-0.5\pm 0.02$ | $-0.91\pm 0.01$ |
>
> These trends are consistent with our existing results. Again, HPI-Clip matches or exceeds performance of all other algorithms, in the sense that its error bars are either higher than, or overlap with, the next-best-performing algorithm.
>
> **[M2: Calculating occupancy-measure-matching (OMM) policies]**
>
> > Is there a memory efficient way to [calculate an OMM policy] without saving all policies?
>
> Yes, this is part of our contribution! Our method does not save intermediate policy iterates, which is a major improvement over SPO. We discussed this briefly on page 6 and will elaborate below.
>
> SPO saves all policy iterates: it re-evaluates policy iterates on validation data to calculate its output policy (see Algorithm 6). This is both memory-intensive and not theoretically justified. In contrast, our method does not save intermediate policies; it saves samples of state-action pairs, and then it applies behaviour cloning with these samples. We show in Lemma 5.4 that this is a theoretically sound way to recover an OMM policy; in the limit of infinitely-many samples from each intermediate policy, an optimally behaviour-cloned policy is an OMM policy.
>
> Algorithms 5 and 6 provide a side-by-side comparison of SPO and HPI-Clip; from there one can see how the return policies differ.
>
> **[M3: Relation to other models of preference]**
>
> The Bradley-Terry (BT) choice model, given by $P((s,a),(s’,a’)) = \sigma(r(s,a) -r(s',a'))$, differs from $\phi$ in the following sense: if $\phi((s,a),(s’,a’)) = \sigma(r(s,a) - r(s’,a’)) -0.5$, then the optimal policy of the Markov decision contest (MDC) with preference model $\phi$ is not necessarily the optimal policy of the MDP with reward function $r$. Mathematically, the issue is that the set of solutions to the game $\max_{\pi} \min_{\pi’} E_{\pi,\pi’}[ \sigma(r(s,a)-r(s’,a’))]$ is not equal to the set of solutions to $ \max_{\pi} E_{\pi}[ r(s,a)]$.
>
> Munos (2024, see Appendices A and B) argue that the pairwise preference model is better than the BT model in many cases. But, of course, there may be cases where the input to the problem is truly stochastic choice data that is sampled from the BT model. Here, the difference is resolved by using the stochastic choice model $P_{\phi}((s,a),(s’,a’)) = \sigma(\phi((s,a),(s’,a’))$ to learn $\phi$ from the data. Under ideal conditions, the learned value for $\phi((s,a),(s’,a’))$ will be $r(s,a) - r(s’,a’)$. And then, by Proposition 4.6, an optimal policy of the MDC with preference function $\phi$ will be an optimal policy of the MDP with reward function $r$.
>
> **[Comments on other weaknesses]**
>
> We'll add evaluation details and a discussion on standard iterative methods for solving MDPs to the main body.
>
> In general, Algorithms 2 and 3 return an OMM policy---we’ll add their return types. In our practical implementation (Algorithm 5), the number of behaviour-cloning steps $N_{BC}$ provides a flexible way to return either the last iterate or approximate OMM policy. When $N_{BC}=0$, the last iterate is returned. When $N_{BC}$ approaches infinity, the return policy is effectively an OMM policy.
>
> Christiano, Paul F., et al. "Deep reinforcement learning from human preferences." Advances in neural information processing systems 30 (2017).
>
> Munos, Rémi, et al. "Nash learning from human feedback." Forty-first International Conference on Machine Learning. 2024.

---

> > ### Author Rebuttal · Reviewer_xuxx · 2026-04-03
> >
> > I thank the authors for their informative rebuttal. I have raised my score.

---

> > > ### Author Response · Authors · 2026-04-06
> > >
> > > We appreciate your consideration. Thank you again for your feedback.

---

### Official Review · Reviewer_XQiw · 2026-03-14

**Soundness:** 2
**Presentation:** 3
**Significance:** 2
**Originality:** 3
**Overall Recommendation:** 4
**Confidence:** 1

**Summary:**

This paper proposed a framework based on Markov decision process to solve the inefficiency of learning the optimal policy based on pairwise preferences. More specifically, they substitute the reward function in MDP with a Markov preference model and formalize it as a model named Markov decision contest. Under the assumption of the transition probability function being unichain and other regularity assumptions, they prove there exists optimal Markov policy for the MDC. They also proposed two algorithms HPI, HPI-CLIP and prove both algorithms converge to the solution in sublinear rate. Simulations with reward-based tasks are provided to support the efficiency of their algorithms.

**Compliance With Llm Reviewing Policy:**

Affirmed.

**Final Justification:**

The rebuttal has addressed my concerns and I decide to increase the score.

**Key Questions For Authors:**

Please refer to strengths and weaknesses for questions.

**Limitations:**

I think this work has no potential negative societal impact.

**Strengths And Weaknesses:**

### Strengths

* This paper is well-written and clearly structured. Assumptions and limitations are made clear in the main text. Theoretical results are supported with proofs and thorough simulations. Connections are made between newly-proposed algorithms and the classical algorithms.

* The idea of formalizing the pairwise preferences learning in the lens of Markov decision process and preference model is innovative and provide a new perspective other than social choice theory and reward-based RL methods.

### Weakness

* Writing: In all algorithms, it should be $1/|\mathcal{A}|$ instead of $1/\mathcal{A}$. In conclusion section, the third line should be "it introduced the Markov decision contest... ". In simulations section, it would be helpful to provide basic information about the reward-based tasks used in the experiments. It would also be helpful to provide further intuitions on why HPI can solve the task (walker2d-nt) that PPO cannot.

* The analysis of the algorithms depends on the assumption of transition probability function being unichain and the fact "mixing time" induced by unichain property. Though it was not made clear that how practical this assumption is and why we need this assumption in obtaining the result.

* In theorem 4.7, the discussion is constrained to $\pi$ - Markov policies while in the comment, it is mentioned that "one that beats or ties every history-dependent policy". It is confusing that whether this optimal Markov policy is also the optimal policy among all history-dependent policies (with well-defined occupancy measures). It would be very helpful if the authors can further explain or rephrase this result.

* Theorem 4.8 is very interesting and surprising, and it would be helpful that how the exponential dependence is resolved and compare the dependence |S||A| with prior works (e.g., PPO, SPO). Would such linear dependence over the state be a problem in practice since |S| is large in practice? e.g. in Example 4.3, the states are the interactions and messages.

* In theorem 5.5, what is the dependence over |S|?

---

> ### Author Rebuttal · Authors · 2026-03-30
>
> Thank you for your feedback. We’re glad you found the paper well written and clearly structured. We’ll first add a clarification to the summary you provided and then respond to other comments. We will include all details from our response and your suggestions in a revised version of the paper.
>
> **[Clarification on paper summary]**
> > Under the assumption of the transition probability function being unichain and other regularity assumptions, they prove there exists optimal Markov policy.
>
> Only the unichain assumption is used—no other regularity assumptions.
>
> **[Performance gap on Walker2d-NT]**
>
> During policy evaluation, SPO (the baseline of Walker2d-NT) averages the per-timestep preferences over the entire trajectory rollout (see Algorithm 6). So, policies may not be evaluated correctly. HPI does not do this averaging, and so we expect it to perform better.
>
> Our ablation study in Appendix G verifies this. We interpolated between SPO and HPI by varying the number of timesteps over which preferences are averaged during policy evaluation. In most (but not all) tasks, averaging leads to worse performance (see Figure 6). Walker2d-NT wasn’t part of the ablation, but its performance gap is likely caused by this.
>
> **[Unichain assumption]**
>
> *Necessity:*
> As discussed in the preliminaries, occupancy measures are not well-defined if there are no assumptions about transition probabilities. The unichain assumption ensures that Markov policies have well-defined occupancy measures, and it is a standard assumption in average-reward MDP analysis. For instance, it’s used in Putterman (2014) and in Sutton and Barto (2018).
>
> *Practicality:* The unichain assumption holds in a number of important cases:
>
> The first case is when contests represent tasks that end at a fixed time horizon. Multi-turn LLM benchmarks like Shani (2024) and robotics tasks, like the ones in our experiments, fall into this case. To formally define the Markov decision contest here, the definition of state must be expanded to include the timestep. But, as we discuss in our response to Reviewer ShcU, this expansion is not unreasonable.
>
> The second case is when contests represent tasks that terminate eventually, but which have variable and unknown time horizons. Mathematically, the assumption is that, with probability one, all states will return to the initial state distribution $\mu$ after a finite amount of time. This seems more realistic than how Shani (2024) models LLM use: people don’t wait until a fixed timestep to start their next conversation. And all conversations end eventually. The distribution $\mu$ can be defined as uniform over the state set, to cover all possible conversations.
>
> Tasks of truly infinite length can also be unichain, provided there is enough connectivity between states.
>
> **[Constraining to Markov policies]**
>
> In Definition 4.4, we defined an optimal Markov policy as one that beats or ties *every history-dependent policy*. So the existence of an optimal Markov policy *can only mean* the existence of a Markov policy that beats or ties every history-dependent policy.
>
> Theorem 4.7 shows that (1) optimal Markov policies exist, and that (2) to find these policies, we only need to compare against other Markov policies, rather than history-dependent ones. So, it justifies constraining to Markov policies. We’ll emphasize this below Theorem 4.7.
>
> **[Theorem 4.8]**
>
> Exponential dependency is resolved by representing the set of all occupancy measures as the set of solutions to a certain system of |S| linear equations. With this representation, the occupancy measure of an optimal policy can be obtained quickly using a linear program (LP). An optimal policy is then recovered from its occupancy measure using Lemma A.1. This is the same approach used to solve average-reward MDPs with linear programming.
>
> We compare our LP with the LP used to solve average-reward MDPs in Appendix B.3. As discussed there, ours has |S||A| more constraints and |S| + 1 more variables.
>
> Lastly, yes: linear dependence is problematic (both for MDPs and MDCs). This is a good point to highlight: it motivates our iterative algorithms.
>
> **[Dependence over |S| in Theorem 5.5]**
>
> The HPI algorithm updates action probabilities at each state in parallel, so its convergence rate (Theorem 5.5) does not explicitly depend on |S|. The convergence rate depends implicitly on |S| through the mixing time $\tau$: if |S| is large, then $\tau$ may also be large, because it may take longer for a policy’s visitation frequency to converge to its occupancy measure.
>
> Please let us know how we can address any concerns that remain.
>
> Puterman, Martin L. Markov decision processes: discrete stochastic dynamic programming. John Wiley & Sons, 2014.
>
> Shani, Lior, et al. "Multi-turn reinforcement learning with preference human feedback." Advances in Neural Information Processing Systems 37 (2024).
>
> Sutton, R. S., & Barto, A. G. (2018). Reinforcement Learning: An Introduction (Second ed.). The MIT Press.

---

> > ### Author Rebuttal · Reviewer_XQiw · 2026-04-04
> >
> > I thank the authors for their thoughtful reply, and hence I would like to increase my score.

---

> > > ### Author Response · Authors · 2026-04-06
> > >
> > > Thank you, we look forward to incorporating your feedback in the revised paper.

---

### Decision · Program_Chairs · 2026-04-30

**Decision:**

Accept (regular)

**Comment:**

This paper formalizes the notion of Markov decision context to study long-term decision making problem with pairwise preferences. Reviewers have appreciated the theoretical framework and the associated algorithm and convergence results. There were some concerns since the experiments were conducted in simulation environments, however, I don't think it is a major concern since the primary contribution of the paper lies in developing the theoretical framework and results. The authors also skillfully answered all the questions raised by the reviewers during the rebuttal. I suggest the authors to improve the presentation of the paper based on the discussion.